# INFORMED POMDP: LEVERAGING ADDITIONAL INFORMATION IN MODEL-BASED RL

## ABSTRACT

In this work, we generalize the problem of learning through interaction in a POMDP by accounting for eventual additional information available at training time. First, we introduce the informed POMDP, a new learning paradigm offering a clear distinction between the training information and the execution observation. Next, we propose an objective that leverages this information for learning a sufficient statistic of the history for the optimal control. We then adapt this informed objective to learn a world model able to sample latent trajectories. Finally, we empirically show a significant learning speed improvement in most environments using this informed world model in the Dreamer algorithm. These results and the simplicity of the proposed adaptation advocate for a systematic consideration of eventual additional information when learning in a POMDP using model-based RL.

## 1 INTRODUCTION

Reinforcement learning (RL) aims to learn to act optimally through interaction with environments whose dynamics are unknown. A major challenge in this field is partial observability, where only a partial observation $o$ of the Markovian state of the environment $s$ is available for taking action $a$. Such an environment can be formalized as a partially observable Markov decision process (POMDP). In this context, an optimal policy $\eta(a|h)$ generally depends on the history $h$ of all observations and previous actions, which grows linearly with time. Fortunately, it is theoretically possible to find a statistic $f(h)$ of the history $h$ that is updated recurrently and that summarizes all relevant information to act optimally. Such a statistic is said to be recurrent and sufficient for the optimal control. Formally, a recurrent statistic is a statistic $f(h)$ updated according to $f(h') = u(f(h), a, o')$ each time an action $a$ is taken and a new observation $o'$ is received, with $h' = (h, a, o')$. A sufficient statistic for the optimal control is a statistic $f(h)$ for which there exists an optimal policy $\eta(a|h) = g(a|f(h))$.

In view of the existence of recurrent and sufficient statistics, many approaches have relied on learning a recurrent policy $\eta_{\theta,\phi}(a|h) = g_\phi(a|f_\theta(h))$ using a recurrent neural network (RNN) $f_\theta$ for the statistic. These policies are simply trained by stochastic gradient ascent of a RL objective using backpropagation through time (Bakker, 2001; Wierstra et al., 2010; Hausknecht & Stone, 2015; Heess et al., 2015; Zhang et al., 2016; Zhu et al., 2017). In this case, the RNN learns a sufficient statistic $f_\theta(h)$ as it learns an optimal policy (Lambrechts et al., 2022; Hennig et al., 2023). Although these approaches theoretically allow implicit learning of a sufficient statistic, sufficient statistics can also be learned explicitly. Notably, many works (Igl et al., 2018; Buesing et al., 2018; Guo et al., 2018; Gregor et al., 2019; Han et al., 2019; Guo et al., 2020; Lee et al., 2020; Hafner et al., 2019; 2020) focused on learning a recurrent statistic that encodes the reward and next observation distribution given the action: $p(r, o'|h, a) = p(r, o'|f(h), a)$, a property known as predictive sufficiency (Bernardo & Smith, 2009). A recurrent and predictive sufficient statistic is indeed proven to be sufficient for the optimal control (Subramanian et al., 2022). The sufficiency objective is usually pursued jointly with the RL objective.

While these methods can learn sufficient statistics and optimal policies in the context of POMDPs, they learn solely from the observations. However, assuming the same partial observability at training time and execution time is too pessimistic for many environments, notably for those that are simulated. We claim that additional information about the state $s$, be it partial or complete, can be leveraged during training for learning sufficient statistics more efficiently. To this end, we generalize the problem of learning from interaction in a POMDP by proposing the informed POMDP. This formalization introduces the training information $i$ about the state $s$, which is only available at training

time. Importantly, this training information is designed such that the observation is conditionally independent of the state given the information. Note that it is always possible to design such an information $i$, possibly by concatenating the observation $o$ with the eventual additional observations $o^+$, such that $i = (o, o^+)$. This formalization offers a new learning paradigm where the training information is used along the reward and observation to supervise the learning of the policy.

In this context, we prove that recurrent statistics are sufficient for the optimal control when they are predictive sufficient for the reward and next information given the action: $p(r, i'|h, a) = p(r, i'|f(h), a)$. We then derive a learning objective for finding a predictive sufficient statistic, which amounts to approximating the conditional distribution $p(r, i'|h, a)$ through likelihood maximization using a model $q_\theta(r, i'|f_\theta(h), a)$, where $f_\theta$ is the statistic. Compared to the classic objective for learning sufficient statistics (Igl et al., 2018; Buesing et al., 2018; Han et al., 2019; Hafner et al., 2019), this objective approximates $p(r, i'|h, a)$ instead of $p(r, o'|h, a)$. Next, we show that this learned model $q_\theta(r, i'|f_\theta(h), a)$ can be adapted to provide a world model from which latent trajectories can be sampled without explicitly reconstructing the observation. This approach boils down to adapting world models such as those of PlaNet or Dreamer (Hafner et al., 2019; 2020; 2021; 2023) by relying on a model of the information instead of a model of the observation. Our claims are supported by experiments in several environments that we formalize as informed POMDPs (Mountain Hike, Velocity Control, Pop Gym, Flickering Atari and Flickering Control). The informed adaptation of Dreamer exhibits a significant improvement in terms of convergence speed and policy performance in most environments, while sometimes hurting performance in others.

This work is structured as follows. In Section 2, we present some related works in asymmetric learning and multi-agent RL. In Section 3, the informed POMDP is presented with the underlying execution POMDP. In Section 4, we provide a learning objective for sufficient statistics in this context. In Section 5, we adapt the Dreamer algorithm to informed POMDPs using this informed objective. In Section 6, we compare the Uninformed Dreamer and the Informed Dreamer in several environments.

## 2 RELATED WORKS

In RL for POMDPs, asymmetric learning consists of exploiting state information during training. These approaches usually learn policies for the POMDP by imitating a policy conditioned on the state (Choudhury et al., 2018). However, these heuristic approaches lack a theoretical framework, and the resulting policies are known to be suboptimal for the POMDP (Warrington et al., 2021; Baisero et al., 2022). Intuitively, optimal policies in POMDP might indeed need to consider actions that reduce the state uncertainty. Warrington et al. (2021) addressed this issue by constraining the expert policy so that its imitation results in an optimal policy in the POMDP. Alternatively, asymmetric actor-critic approaches use a critic conditioned on the state (Pinto et al., 2018). These approaches have been proven to provide biased gradients by Baisero & Amato (2022), who also proposed an unbiased actor-critic approach by introducing the history-state value function $V(h, s)$. Baisero et al. (2022) adapted this method to value-based RL, where the history-dependent value function $V(h)$ uses the history-state value function $V(h, s)$ in its temporal difference target. Alternatively, Nguyen et al. (2021) proposed to enforce that the statistic $f(h)$ encodes the belief $p(s|h)$, a sufficient statistic for the optimal control (Åström, 1965). It requires making the strong assumption that beliefs $b(s) = p(s|h)$ are available at training time. Finally, in the work that is the closest to ours, Avalos et al. (2023) learns a statistic $f(h)$ that encodes the belief distribution $p(s|h)$ by leveraging the states during training.

In multi-agent RL, exploiting additional information available at training time was extensively studied under the centralized training and decentralized execution (CTDE) framework (Oliehoek et al., 2008). In CTDE, it is assumed that the histories of all agents, or even the environment state, are available to all agents at training time. To exploit this additional information, several asymmetric actor-critic approaches have been developed by leveraging an asymmetric critic conditioned on all histories, including COMA (Foerster et al., 2018), MADDPG (Lowe et al., 2017), M3DDPG (Li et al., 2019) and R-MADDPG (Wang et al., 2020). While efficient in practice, Lyu et al. (2022) showed that these asymmetric actor-critic approaches provide biased gradient estimates, which generalizes results developed for asymmetric learning in POMDP (Baisero & Amato, 2022) to the multi-agent setting. In the cooperative CTDE setting, another line of work focuses on value decomposition to learn a utility function for each agent, including QMIX (Rashid et al., 2018), QVMix (Leroy et al., 2021) and QPLEX (Wang et al., 2021). These approaches use the additional information to modulate the

contribution of each utility function in the global value function, while ensuring that maximizing the local utility functions also maximize the global value function, a property known as individual global max (IGM). Other methods relax this IGM requirement but still condition the value function on all histories, including QTRAN (Son et al., 2019) and WQMix (Rashid et al., 2020). Recently, Hong et al. (2022) established that the IGM decomposition is not attainable in the general case.

In contrast to the existing literature on asymmetric learning in POMDP, we introduce an approach that is guaranteed to provide a sufficient statistic for the optimal control, and that leverages the additional information only through the objective. Moreover, our new learning paradigm is not restricted to state supervision, but supports any level of additional information. Finally, to the best of our knowledge, our method is the first to exploit additional information for learning an environment model of the POMDP. While our approach is probably applicable to the CTDE setting for learning sufficient statistics from the local histories of each agent, we leave it as future work.

## 3 INFORMED POMDP

In this section, we introduce the informed POMDP and the associated training information, along with the underlying execution POMDP and the RL objective in this context.

### 3.1 INFORMED POMDP AND EXECUTION POMDP

Formally, an informed POMDP $\widetilde{\mathcal{P}}$ is defined as a tuple $\widetilde{\mathcal{P}} = (\mathcal{S}, \mathcal{A}, \mathcal{I}, \mathcal{O}, T, R, \widetilde{I}, \widetilde{O}, P, \gamma)$ where $\mathcal{S}$ is the state space, $\mathcal{A}$ is the action space, $\mathcal{I}$ is the information space, and $\mathcal{O}$ is the observation space. The initial state distribution $P$ gives the probability $P(s_0)$ of $s_0 \in \mathcal{S}$ being the initial state of the decision process. The dynamics are described by the transition distribution $T$ that gives the probability $T(s_{t+1}|s_t, a_t)$ of $s_{t+1} \in \mathcal{S}$ being the state resulting from action $a_t \in \mathcal{A}$ in state $s_t \in \mathcal{S}$. The reward function $R$ gives

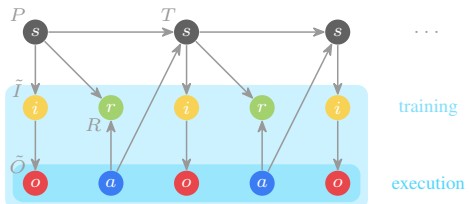

Figure 1: Bayesian network of an informed POMDP execution.

the expected immediate reward $r_t = R(s_t, a_t)$ obtained at each transition. The information distribution $\widetilde{I}$ gives the probability $\widetilde{I}(i_t|s_t)$ to get information $i_t \in \mathcal{I}$ in state $s_t \in \mathcal{S}$, and the observation distribution $\widetilde{O}$ gives the probability $\widetilde{O}(o_t|i_t)$ to get observation $o_t \in \mathcal{O}$ given information $i_t$. Finally, the discount factor $\gamma \in [0, 1]$ gives the relative importance of future rewards. The main assumption about an informed POMDP is that the observation $o_t$ is conditionally independent of the state $s_t$ given the information $i_t$: $p(o_t|i_t, s_t) = \widetilde{O}(o_t|i_t)$. In other words, the random variables $s_t$, $i_t$ and $o_t$ satisfy the Bayesian network $s_t \longrightarrow i_t \longrightarrow o_t$. In practice, it is always possible to define such a training information $i_t$. For example, the information $i_t = (o_t, o_t^+)$ satisfies the aforementioned conditional independence for any $o_t^+$. Taking a sequence of $t$ actions in the informed POMDP conditions its execution and provides samples $(i_0, o_0, a_0, r_0, \ldots, i_t, o_t)$ at training time, as illustrated in Figure 1.

For each informed POMDP, there is an underlying execution POMDP that is defined as $\mathcal{P} = (\mathcal{S}, \mathcal{A}, \mathcal{O}, T, R, O, P, \gamma)$, where $O(o_t|s_t) = \int_{\mathcal{I}} \widetilde{O}(o_t|i)\widetilde{I}(i|s_t) \, \mathrm{d}i$. Taking a sequence of $t$ actions in the execution POMDP conditions its execution and provides the history $h_t = (o_0, a_0, \ldots, o_t) \in \mathcal{H}$, where $\mathcal{H}$ is the set of histories of arbitrary length. Note that the information samples $i_0, \ldots, i_t$ and reward samples $r_0, \ldots, r_{t-1}$ are not included, since they are not available at execution time.

### 3.2 RL OBJECTIVE

A policy $\eta \in H$ is a mapping from histories to probability measures over the action space, where $H = \mathcal{H} \to \Delta(\mathcal{A})$ is the set of such mappings. A policy is said to be optimal for an informed POMDP when it is optimal in the underlying execution POMDP, i.e., when it maximizes the expected return,

$$J(\eta) = \mathbb{E}_{\substack{P(s_0) \\ O(o_t|s_t) \\ \eta(a_t|h_t) \\ T(s_{t+1}|s_t, a_t)}} \left[ \sum_{t=0}^{\infty} \gamma^t R(s_t, a_t) \right]. \tag{1}$$

The RL objective for an informed POMDP is thus to find an optimal policy $\eta^* \in \arg\max_{\eta \in H} J(\eta)$ for the execution POMDP from interaction with the informed POMDP.

# 4 OPTIMAL CONTROL WITH RECURRENT SUFFICIENT STATISTICS

In this section, we introduce the notion of sufficient statistic for the optimal control and derive an objective for learning such a statistic in an informed POMDP. For the sake of conciseness, we simply use $x$ to denote a random variable at the current time step and $x'$ to denote it at the next time step. Moreover, we use the composition notation $g \circ f$ to denote the history-dependent policy $g(\cdot|f(\cdot))$.

## 4.1 RECURRENT SUFFICIENT STATISTICS

Let us first define the concept of sufficient statistic, and derive a necessary condition for optimality.

**Definition 1** (Sufficient statistic). In an informed POMDP $\widetilde{\mathcal{P}}$ and in its underlying execution POMDP $\mathcal{P}$, a statistic of the history $f \colon \mathcal{H} \to \mathcal{Z}$ is sufficient for the optimal control if, and only if,

$$\max_{g \colon \mathcal{Z} \to \Delta(\mathcal{A})} J(g \circ f) = \max_{\eta \colon \mathcal{H} \to \Delta(\mathcal{A})} J(\eta). \tag{2}$$

**Corollary 1** (Sufficiency of optimal policies). In an informed POMDP $\widetilde{\mathcal{P}}$ and in its underlying execution POMDP $\mathcal{P}$, if a policy $\eta = g \circ f$ is optimal, then the statistic $f : \mathcal{H} \to \mathcal{Z}$ is sufficient for the optimal control.

In this work, we focus on learning recurrent policies, i.e., policies $\eta = g \circ f$ for which the statistic $f$ is recurrent. Formally, we have,

$$\eta(a|h) = g(a|f(h)), \ \forall(h, a), \tag{3}$$

$$f(h') = u(f(h), a, o'), \ \forall h' = (h, a, o'). \tag{4}$$

This enables the history to be processed iteratively each time that an action is taken and an observation is received. According to Corollary 1, when learning a recurrent policy $\eta = g \circ f$, the objective can be broken down into two problems: finding a sufficient statistic $f$ and an optimal distribution $g$,

$$\max_{\substack{f \colon \mathcal{H} \to \mathcal{Z} \\ g \colon \mathcal{Z} \to \Delta(\mathcal{A})}} J(g \circ f). \tag{5}$$

## 4.2 LEARNING RECURRENT SUFFICIENT STATISTICS

Below, we provide a sufficient condition for a statistic to be sufficient for the optimal control.

**Theorem 1** (Sufficiency of recurrent predictive sufficient statistics). In an informed POMDP $\widetilde{\mathcal{P}}$, a statistic $f \colon \mathcal{H} \to \mathcal{Z}$ is sufficient for the optimal control if it is (i) recurrent and (ii) predictive sufficient for the reward and next information given the action,

$$\text{(i)} \ f(h') = u(f(h), a, o'), \ \forall h' = (h, a, o'), \tag{6}$$

$$\text{(ii)} \ p(r, i'|h, a) = p(r, i'|f(h), a), \ \forall(h, a, r, i'). \tag{7}$$

The proof for this theorem is in Appendix A, generalizing earlier work by Subramanian et al. (2022).

Now, let us consider a distribution over the histories and actions whose density function is denoted as $p(h, a)$. For example, we consider the stationary distribution induced by the current policy $\eta$ in the informed POMDP $\widetilde{\mathcal{P}}$. Let us also assume that the density function $p(h, a)$ is non-zero everywhere. As shown in Appendix B, under mild assumption, any statistic $f_\theta$ satisfying the following objective,

$$\max_{\substack{f \colon \mathcal{H} \to \mathcal{Z} \\ q \colon \mathcal{Z} \times \mathcal{A} \to \Delta(\mathbb{R} \times \mathcal{I})}} \mathbb{E}_{p(h, a, r, i')} \log q(r, i'|f(h), a), \tag{8}$$

also satisfies (ii). This variational objective jointly optimizes the statistic function $f : \mathcal{H} \to \mathcal{Z}$ with a conditional probability density function $q \colon \mathcal{Z} \times \mathcal{A} \to \Delta(\mathbb{R} \times \mathcal{I})$. According to Theorem 1, a statistic that is recurrent and that satisfies objective (8) is sufficient for the optimal control.

In practice, both the recurrent statistic and the density function are implemented with neural networks $f_\theta$ and $q_\theta$ respectively, both parametrized by $\theta \in \mathbb{R}^d$. In this case, the objective can be maximized by stochastic gradient ascent. Regarding the statistic function $f_\theta$, it is implicitly implemented by the update function $z_t = u_\theta(z_{t-1}; x_t)$ of an RNN. The inputs are $x_t = (a_{t-1}, o_t)$, with $a_{-1}$ the null action that is typically set to zero. The hidden state of the RNN $z_t = f_\theta(h_t)$ is thus a statistic of the history that is recurrently updated using $u_\theta$. Regarding $q_\theta$, it is implemented by a parametrized probability density function estimator. In such a context, we obtain the following objective,

$$\max_\theta \underbrace{\mathbb{E}_{p(h,a,r,i')} \log q_\theta(r, i'|f_\theta(h), a)}_{L(f_\theta)}. \tag{9}$$

We might wonder whether this informed objective is better than the classic objective, where $i = o$. In this work, we hypothesize that regressing the information distribution instead of the observation distribution is a better objective in practice. This is motivated by the data processing inequality applied to the Bayesian network $s' \longrightarrow i' \longrightarrow o'$, which concludes that the information $i'$ is more informative than the observation $o'$ about the Markovian state $s'$ of the environment,

$$I(s', i'|h, a) \geq I(s', o'|h, a). \tag{10}$$

where $I$ denotes the conditional mutual information. We thus expect the statistic $f_\theta(h)$ to converge faster towards a sufficient statistic, and the policy to converge faster towards an optimal policy. It is however important to note that the information $i$ might contain irrelevant or exogenous state variables. In practice, the conditional distribution $p(i'|h, a)$ may thus be much more difficult to approximate than $p(o'|h, a)$, while not being much more useful to the control task. While we consider this study out of the scope of this work, ensuring that the sufficient representations of the histories are also necessary for the control task is a promising avenue for future work.

### 4.3 OPTIMAL CONTROL WITH RECURRENT SUFFICIENT STATISTICS

As seen from Corollary 1, sufficient statistics are needed for the optimal control of POMDPs. Moreover, as we focus on recurrent policies implemented with RNNs, we can exploit objective (9) to learn a sufficient statistic $f_\theta$. In practice, we jointly maximize the RL objective $J(\eta_{\theta,\phi}) = J(g_\phi \circ f_\theta)$ and the statistic objective $L(f_\theta)$. This enables one to use the information $i$ to guide the statistic learning through $L(f_\theta)$. This joint maximization results in the following objective,

$$\max_{\theta,\phi} J(g_\phi \circ f_\theta) + L(f_\theta). \tag{11}$$

Note that a policy maximizing (11) also maximizes the return $J(g_\phi \circ f_\theta)$ if $f_\theta$ and $q_\theta$ are expressive enough, such that this objective provides optimal policies in the sense of objective (5).

## 5 MODEL-BASED RL THROUGH INFORMED WORLD MODELS

Model-based RL focuses on learning a model of the dynamics $p(r, o'|h, a)$ of the environment, known as a world model, that is exploited to derive a near-optimal policy. Since the approximate model usually allows one to generate trajectories, many works derive a near-optimal policy by online planning (e.g., model-predictive control) or by optimizing a parametrized policy based on these trajectories (Sutton, 1991; Ha & Schmidhuber, 2018; Chua et al., 2018; Zhang et al., 2019; Hafner et al., 2019; 2020). In this section, we first modify the model $q_\theta(r, i'|f_\theta(h), a)$ in order to get a world model from which trajectories can be sampled. We then adapt the DreamerV3 (Hafner et al., 2023) algorithm using this world model, resulting in the Informed Dreamer algorithm.

### 5.1 INFORMED WORLD MODEL

In this work, we implement the informed world model with a variational RNN (VRNN) as introduced by Chung et al. (2015), also known as a recurrent state-space model (RSSM) in the RL context (Hafner et al., 2019). It is worth noticing that such a model performs its recurrent update using a latent stochastic representation of the observation. When generating trajectories, it also samples latent representations of the observations without explicitly reconstructing them, which we refer to as latent

trajectories. This key design choice enables the sampling of trajectories without explicitly learning the observation distribution, but the reward and information distribution only. Formally, we have,

$$\hat{e} \sim q_\theta^p(\cdot|z,a), \qquad \text{(prior, 12)}$$

$$\hat{r} \sim q_\theta^r(\cdot|z,\hat{e}), \qquad \text{(reward decoder, 13)}$$

$$\hat{i}' \sim q_\theta^i(\cdot|z,\hat{e}), \qquad \text{(information decoder, 14)}$$

where $\hat{e}$ is the latent variable of the VRNN when generating trajectories. The prior $q_\theta^p$ and the decoders $q_\theta^i$ and $q_\theta^r$ are jointly trained with the encoder,

$$e \sim q_\theta^e(\cdot|z,a,o'), \qquad \text{(encoder, 15)}$$

to maximize the likelihood of reward and next information samples. The latent representation $e \sim q_\theta^e(\cdot|z,a,o')$ of the next observation $o'$ can be used to update the statistic to $z'$,

$$z' = u_\theta(z,a,e). \qquad \text{(recurrence, 16)}$$

Note that the statistic $z$ is no longer deterministically updated to $z'$ given $a$ and $o'$, instead we have $z \sim f_\theta(\cdot|h)$, which is induced by $u_\theta$ and $q_\theta^e$. In practice, we maximize the evidence lower bound (ELBO), a variational lower bound on the likelihood of reward and next information samples given the statistic (Chung et al., 2015),

$$\mathbb{E}_{\substack{p(h,a,r,i') \\ f_\theta(z|h)}} \log q_\theta(r,i'|z,a) \geq \mathbb{E}_{\substack{p(h,a,r,i',o') \\ f_\theta(z|h)}} \left[ \mathbb{E}_{q_\theta^e(e|z,a,o')} \left[ \log q_\theta^i(i'|z,e) + \log q_\theta^r(r|z,e) \right] \right.$$
$$\left. - \text{KL}\left( q_\theta^e(\cdot|z,a,o') \parallel q_\theta^p(\cdot|z,a) \right) \right]. \qquad (17)$$

As illustrated in Figure 2 for a trajectory sampled in the informed POMDP, the ELBO maximizes the conditional log-likelihood $q_\theta^r(r|z,e)$ and $q_\theta^i(i|z,e)$ of $r$ and $i'$ for a sample of the encoder $e \sim q_\theta^e(\cdot|z,a,o')$, and minimizes the KL divergence from $q_\theta^e(\cdot|z,a,o')$ to the prior distribution $q_\theta^p(\cdot|z,a)$. Note that when $i = o$, it corresponds to Dreamer's world model and learning objective.

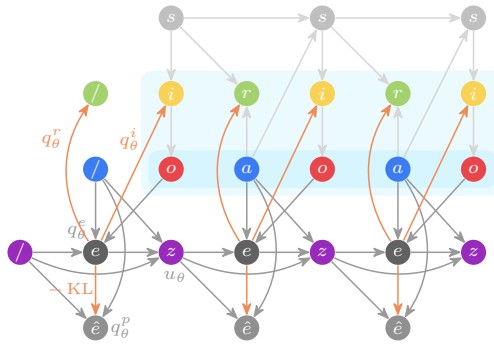

Figure 2: VRNN loss for a given trajectory at training time. Dependence of $q_\theta^r$ and $q_\theta^i$ on $z$ is omitted.

As can be noticed from Equation 17 and Figure 2, the encoder is conditioned on the observation only. While this is required for the encoder to be used at execution time, it certainly loosen the lower bound and limits the quality of the conditional information distribution that can be learned. Future work may improve the quality of the information reconstruction by considering an additional information encoder, also conditioned on the statistic of the history, whose samples are not used in the recurrence.

## 5.2 INFORMED DREAMER

As explained above, while our informed world model does not learn the observation distribution, it is still able to sample latent trajectories. Indeed, the VRNN only uses the latent representation $e \sim q_\theta^e(\cdot|z,a,o')$ of the observation $o'$, trained to reconstruct the information $i'$, in order to update $z$ to $z'$. Consequently, we can use the prior distribution $\hat{e} \sim q_\theta^p(\cdot|z,a)$, trained according to (17) to minimize the KL divergence from $e \sim q_\theta^e(\cdot|z,a,o')$ in expectation, to sample latent trajectories.

The Informed Dreamer algorithm leverages such trajectories to learn a latent critic $v_\psi(z)$ and a latent policy $a \sim g_\phi(\cdot|z)$. Figure 3a illustrates the generation of a latent trajectory, along with estimated

rewards $\hat{r} \sim q_\theta^r(\cdot|z, e)$ and values $\hat{v} = v_\psi(z)$. The actions are sampled according to the latent policy, and any RL algorithm can be used to maximize the estimated return. Moreover, note that the estimated return is given by a function that is differentiable with respect to $\phi$, and it can be directly maximized by stochastic gradient ascent. In the experiments, we use an actor-critic approach for discrete actions and direct maximization for continuous actions, following DreamerV3 (Hafner et al., 2023). Finally, as shown in Figure 3b, when deployed in the execution POMDP, the encoder $q_\theta^e$ is used to compute the latent representations of the observations and to update the statistic. The actions are then selected according to $a \sim g_\phi(\cdot|z)$.

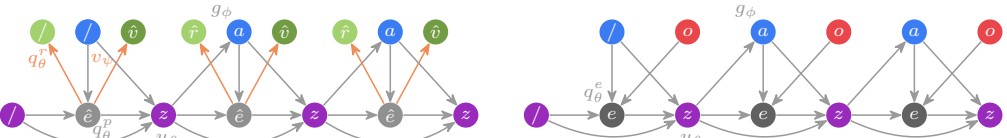

(a) Imagination of a trajectory using policy $g_\phi$ with estimated rewards and values. Dependence of $q_\theta^r$ and $v_\psi$ on $z$ is omitted.

(b) Execution of the policy on a trajectory of the POMDP using the encoder $q_\theta^e$ to condition the latent policy $g_\phi$.

Figure 3: Bayesian graph of a VRNN evaluation during imagination and execution.

A pseudocode for the adaptation of the DreamerV3 algorithm using this informed world model is given in Appendix C. We also detail some divergences of our formalization with respect to the original DreamerV3 algorithm. As in DreamerV3, we use symlog predictions, a discrete VAE, KL balancing, free bits, reward normalisation, a distributional critic, and entropy regularization.

## 6 EXPERIMENTS

In this section, we compare Dreamer to the Informed Dreamer on several informed POMDPs, all considered with a discount factor of $\gamma = 0.997$. For reproducibility purposes, we use the implementation and hyperparameters of DreamerV3 released by the authors at github.com/danijar/dreamerv3, and release our adaptation to informed POMDPs using the same hyperparameters at [anonymized].

### 6.1 VARYING MOUNTAIN HIKE

In the Varying Mountain Hike environments, the agent should walk throughout a mountainous terrain to reach the mountain top as fast as possible while avoiding the valleys. There exists four versions of this environment, depending on the agent orientation (*north* or *random*) and on the observation that is available (*position* or *altitude*). More formally, the agent has a position $x$ and a fixed orientation $c$ in each episode. The orientation $c$ is either always *north* or a *random* cardinal orientation, depending on the environment version. It can take four actions to move relative to its orientation (right, forward, left and backward). The orientation is not observed by the agent, but it receives a Gaussian observation of its *position*, or its *altitude*, depending on the environment version ($\sigma_o = 0.1$ in both cases). The reward is given by its altitude relative to the mountain top, such that the goal of the agent is to obtain the highest cumulative altitude. Around the mountain top, states are terminal and the trajectories are truncated at $t = 160$ in practice. We refer the reader to Lambrechts et al. (2022) for a formal description of these environments, strongly inspired by the Mountain Hike of Igl et al. (2018).

For this environment, we first consider the position and orientation to be available as additional information at training time. In other words, we consider the state-informed POMDP with $i = s$. As can be seen in Figure 4a, the speed of convergence of the policies is improved in all four environments when using the Informed Dreamer. Moreover, as shown in Table 1 in Appendix D, the final performance of the Informed Dreamer is better in 3 out of 4 environments.

We also experiment with other types of information in the Varying Mountain Hike with position observation and random orientation. More precisely, we consider an information $i = (\tilde{x}, \tilde{c})$ about the state $s = (x, c)$, where $\tilde{x}$ is an observation of the position $x$ with Gaussian noise of standard deviation $\sigma_i \in [0, \sigma_o]$, and $\tilde{c}$ is a noisy observation of the orientation $c$ replaced by a random orientation with probability $\epsilon_i \in [0, 1]$. Note that when $\sigma_i = 0$, the position $x$ is encoded in the information, while when $\sigma_i = \sigma_o$, the observation $o$ is encoded in the information. As shown in Figure 4b, without confidence intervals for the sake readability, the better the information, the faster the policy converges. It supports the idea that the more information about the state is exploited, the faster an optimal

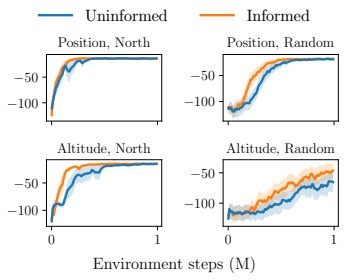
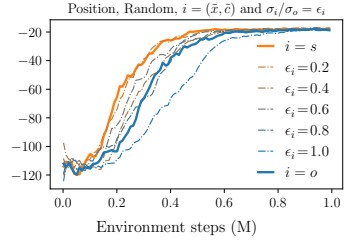

(a) Uninformed Dreamer and Informed Dreamer with $i = s$ in the four environments.

(b) Informed Dreamer with $i = (\tilde{x}, \tilde{c})$ with position observation and random orientation.

Figure 4: Varying Mountain Hike environments: average return and standard error over five trainings.

policy for the POMDP is learned. Moreover, we observe that the Informed Dreamer with $\epsilon_i = 1$ and $\sigma_i = 0.1$ performs even worse than the Uninformed Dreamer. It suggests that considering additional information that is not informative about the state (i.e., $I(s, i|o) = 0$), such as $\tilde{c}$ with $\epsilon_i = 1$, can degrade learning. Similar results are obtained for the other three environments in Subsection E.1.

## 6.2 VELOCITY CONTROL

In the Velocity Control environments, we consider the standard DeepMind Control tasks (Tassa et al., 2018), where only the joints velocities are available as observations and not their absolute positions, which is a standard benchmark in the partially observable RL literature (Han et al., 2019; Lee et al., 2020). These environments consists of controlling different multi-joints robots to achieve several tasks. We consider the absolute positions to be available at training time along with the velocities, which results in a Markovian information $i = s$.

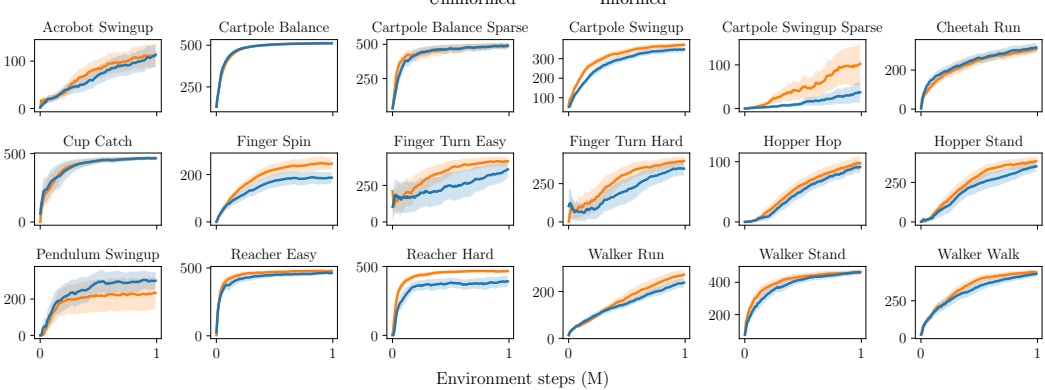

Figure 5: Uninformed Dreamer and Informed Dreamer with $i = s$ in the Velocity Control environments: average return and standard error over five trainings.

Figure 5 shows that the convergence speed of the policies is greatly improved in this benchmark, for nearly all of the considered games. Moreover, the final returns are given in Table 2 in Appendix D, and show that policies obtained after one million time steps are better in 13 out of 18 environments when considering additional information.

## 6.3 POP GYM

The Pop Gym environments have been specifically designed to benchmark the ability of handling partial observability (Morad et al., 2023). The latter notably includes memory games, board games, or control problems involving partial observability and noise. For these environments, we consider the state to be available as additional information.

Figure 6 shows that learning in those POMDPs usually benefits from the exploitation of additional information as proposed in the Informed Dreamer. The learning of the Informed Dreamer seems to suffer from the approximation of the information distribution in only 2 out of those 10 environments (Concentration and Higher Lower). The final returns are given in Table 3 in Appendix D,

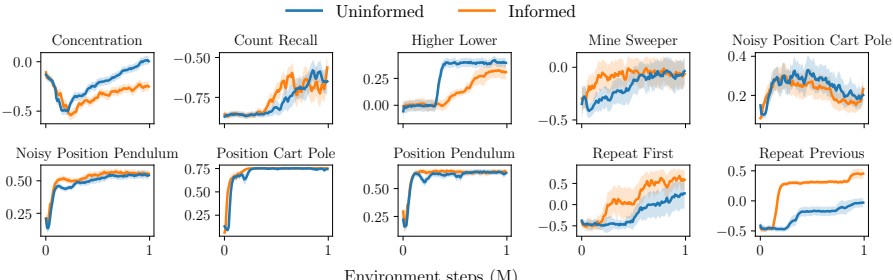

Figure 6: Uninformed Dreamer and Informed Dreamer with $i = s$ in the Pop Gym environments: average return and standard error over five trainings.

showing a better final performance in 7 out of 10 environments. In particular, like in the Velocity Control environments, we observe for the Cart Pole and Pendulum environments that the Informed Dreamer converges to a better performance. This also holds for the Repeat First and Repeat Previous environments, that both require discovering long time dependencies. The exploitation of additional information seems crucial in these environments, and we study this in depth for the Repeat Previous environment in Subsection E.2. This analysis shows that the Informed Dreamer can learn near-optimal policies in environments for which the Uninformed Dreamer does not learn at all.

### 6.4 FLICKERING ATARI AND FLICKERING CONTROL

While arguably not constituting a relevant benchmark for measuring the ability of handling partial observability (Shao et al., 2022; Avalos et al., 2023), the Flickering Atari and Flickering Control environments have become standard benchmarks in the partially observable RL literature (Hausknecht & Stone, 2015; Zhu et al., 2017; Igl et al., 2018; Ma et al., 2020). For completeness, the results for these environments are reported in Appendix E. We observe that the speed of convergence and final performance of the agent is sometimes greatly improved when considering additional information (e.g., Asteroids, Pong, Breakout). However, we also observe that the performance is lower in some environments. As far as the Flickering Atari environments are concerned, the Informed Dreamer only outperforms Dreamer in 6 out of 12 environments. In the Flickering Control environments, the Informed Dreamer tends to systematically underperform the Uniformed Dreamer, attaining a better performance in only 2 out of 18 environments. It suggests that exploiting additional state information is not useful in these environments. We hypothesize that the conditional information distribution is difficult to approximate, which may cause learning to degrade. This shows that not all information is worth exploiting, particularly when the level of partial observability is low.

### 7 CONCLUSION

In this work, we introduced a new formalization for considering additional information available at training time for POMDP, called the informed POMDP. In this context, we proposed an objective for learning recurrent sufficient statistic for the optimal control. Next, we adapted this objective to provide an environment model from which latent trajectories can be sampled. We then adapted a successful model-based RL algorithm, known as Dreamer, with this informed world model, resulting in the Informed Dreamer algorithm. By considering several environments from the partially observable RL literature, we showed that this informed learning objective improves the convergence speed and quality of the policies in most cases. Given the similarities with the CTDE context, this work motivates future work and lays the foundations for developing multi-agent methods that learn sufficient statistics from the local history of each agent. This work also presents several limitations. First, a formal theoretical justification for the use of the information instead of the observation is still lacking. One solution might be to consider the notion of approximate information states to bound the suboptimality of the policy for a given error on the information distribution instead of the observation distribution. Second, we observed that this informed objective hurts performance in some environments, motivating further work in which particular attention is paid to the design of the information. In particular, it would be worth drawing connection to the exogenous RL literature that would complement this work by focusing on discarding irrelevant information. Third, the ELBO learning objective is probably a loose lower bound on the information likelihood, and future works might improve the quality of the information distribution by considering informed world models with two encoders.

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

## A    Sufficiency of recurrent predictive sufficient statistics

In this section, we prove Theorem 1, that is recalled below.

**Theorem 1** (Sufficiency of recurrent predictive sufficient statistics). *In an informed POMDP $\widetilde{\mathcal{P}}$, a statistic $f\colon \mathcal{H} \to \mathcal{Z}$ is sufficient for the optimal control if it is (i) recurrent and (ii) predictive sufficient for the reward and next information given the action,*

$$\text{(i) } f(h') = u(f(h), a, o'), \ \forall h' = (h, a, o'), \tag{6}$$

$$\text{(ii) } p(r, i'|h, a) = p(r, i'|f(h), a), \ \forall (h, a, r, i'). \tag{7}$$

*Proof.* From Proposition 4 and Theorem 5 by Subramanian et al. (2022), we know that a statistic is sufficient for the optimal control of an execution POMDP if it is (i) recurrent and (ii') predictive sufficient for the reward and next *observation* given the action: $p(r, o'|h, a) = p(r, o'|f(h), a)$. Let us consider a statistic $f\colon \mathcal{H} \to \mathcal{A}$ satisfying (i) and (ii). Let us show that it satisfies (ii'). We have,

$$p(r, o'|f(h), a) = \int_{\mathcal{I}} p(r, o', i'|f(h), a) \, \mathrm{d}i' \tag{18}$$

$$= \int_{\mathcal{I}} p(o'|r, i', f(h), a) p(r, i'|f(h), a) \, \mathrm{d}i', \tag{19}$$

using the law of total probability and the chain rule. As can be seen from the informed POMDP formalization of Section 3 and the resulting Bayesian network in Figure 1, the Markov blanket of $o'$ is $\{i'\}$. As a consequence, $o'$ is conditionally independent of any other variable given $i'$. In particular, $p(o'|i', r, f(h), a) = p(o|i')$, such that,

$$p(r, o'|f(h), a) = \int_{\mathcal{I}} p(o'|i') p(r, i'|f(h), a) \, \mathrm{d}i'. \tag{20}$$

From hypothesis (ii), we can write,

$$p(r, o'|f(h), a) = \int_{\mathcal{I}} p(o'|i') p(r, i'|h, a) \, \mathrm{d}i'. \tag{21}$$

Finally, exploiting the Markov blanket $\{i'\}$ of $o'$, the chain rule and the law of total probability again, we have,

$$p(r, o'|f(h), a) = \int_{\mathcal{I}} p(o'|i', r, h, a) p(r, i'|h, a) \, \mathrm{d}i' \tag{22}$$

$$= \int_{\mathcal{I}} p(o', r, i'|h, a) \, \mathrm{d}i' \tag{23}$$

$$= p(r, o'|h, a). \tag{24}$$

This proves that (ii) implies (ii'). As a consequence, any statistic $f_\theta$ satisfying (i) and (ii) is a sufficient statistic of the history for the optimal control of the informed POMDP. $\qquad\square$

## B    Recurrent sufficient statistic objective

First, let us consider a fixed history $h$ and action $a$. Let us recall that two density functions $p(r, i'|h, a)$ and $p(r, i'|f(h), a)$ are equal almost everywhere if, and only if, their KL divergence is zero,

$$\underset{p(r,i'|h,a)}{\mathbb{E}} \log \frac{p(r, i'|h, a)}{p(r, i'|f(h), a)} = 0. \tag{25}$$

Now, let us consider a probability density function $p(h, a)$ that is non zero everywhere. We have that the KL divergence from $p(r, i'|h, a)$ to $p(r, i'|f(h), a)$ is equal to zero for almost every history $h$ and action $a$ if, and only if, it is zero on expectation over $p(h, a)$, since the KL divergence is non-negative,

$$\underset{p(r,i'|h,a)}{\mathbb{E}} \log \frac{p(r, i'|h, a)}{p(r, i'|f(h), a)} \overset{\text{a.e.}}{=} 0 \Leftrightarrow \underset{p(h,a,r,i')}{\mathbb{E}} \log \frac{p(r, i'|h, a)}{p(r, i'|f(h), a)} = 0. \tag{26}$$

Rearranging, we have that $p(r, i'|h, a)$ is equal to $p(r, i'|f(h), a)$ for almost every $h$, $a$, $r$ and $i'$ if, and only if,

$$\mathbb{E}_{p(h,a,r,i')} \log p(r, i'|h, a) = \mathbb{E}_{p(h,a,r,i')} \log p(r, i'|f(h), a). \tag{27}$$

Now, we recall the data processing inequality, enabling one to write, for any statistic $f'$,

$$\mathbb{E}_{p(h,a,r,i')} \log p(r, i'|h, a) \geq \mathbb{E}_{p(h,a,r,i')} \log p(r, i'|f'(h), a). \tag{28}$$

since $h(r, i'|h, a) = h(r, i'|h, f(h), a) \leq h(r, i'|f(h), a)$, $\forall (h, a)$, where $h(x)$ is the differential entropy of random variable $x$. Assuming that there exists at least one $f \colon \mathcal{H} \to \mathcal{Z}$ for which the inequality is tight, we obtain the following objective for a predictive sufficient statistic $f$,

$$\max_{f \colon \mathcal{H} \to \mathcal{Z}} \mathbb{E}_{p(h,a,r,i')} \log p(r, i'|f(h), a). \tag{29}$$

Unfortunately, the probability density $p(r, i'|f(h), a)$ is unknown. However, knowing that the distribution that maximizes the log-likelihood of samples from $p(r, i'|f(h), a)$ is $p(r, i'|f(h), a)$ itself, we can write,

$$\mathbb{E}_{p(h,a,r,i')} \log p(r, i'|f(h), a) = \max_{q \colon \mathcal{Z} \times \mathcal{A} \to \Delta(\mathbb{R} \times \mathcal{I})} \mathbb{E}_{p(h,a,r,i')} \log q(r, i'|f(h), a). \tag{30}$$

By jointly maximizing the probability density function $q \colon \mathcal{Z} \times \mathcal{A} \to \Delta(\mathbb{R} \times \mathcal{I})$, we obtain,

$$\max_{\substack{f \colon \mathcal{H} \to \mathcal{Z} \\ q \colon \mathcal{Z} \times \mathcal{A} \to \Delta(\mathbb{R} \times \mathcal{I})}} \mathbb{E}_{p(h,a,r,i')} \log q(r, i'|f(h), a). \tag{31}$$

This objective ensures that the statistic $f(h)$ is predictive sufficient for the reward and next information given the action. If $f(h)$ is a recurrent statistic, then it is also sufficient for the optimal control, according to Theorem 1.

## C  INFORMED DREAMER

The Informed Dreamer algorithm is presented in Algorithm 3. Differences with the Uninformed Dreamer algorithm (Hafner et al., 2020) are highlighted in blue. In addition, it can be noted that in the original Dreamer algorithm, the statistic $z_t$ encodes $h_t = (o_0, a_0, \ldots, o_t)$ and $a_t$, instead of $h_t$ only. As a consequence, the prior distribution $e_t \sim q_\theta^p(\cdot|z_t)$ can be conditioned on the statistic $z_t$ only, instead of the statistic and last action. Similarly, the encoder distribution $e_t \sim q_\theta^p(\cdot|z_t, o_{t+1})$ can be conditioned on the statistic $z_t$ only, instead of the statistic and last action. On the other hand, the latent policy $a_{t+1} \sim g(\cdot|z_t, e_t)$ should be conditioned on the statistic $z_t$ and the new latent $e_t$ to account for the last observation, and the same is true for the value function $v_\psi(z_t, e_t)$. In the experiments, we follow the original implementation for both the Uninformed Dreamer and the Informed Dreamer, according to the code that we release at [anonymized].

Following Dreamer, the algorithm introduces the continuation flag $c_t$, which indicates whether state $s_t$ is terminal. A terminal state $s_t$ is a state from which the agent can never escape, and in which any further action provides a zero reward. It follows that the value function of a terminal state is zero, and trajectories can be truncated at terminal states since we do not need to learn their value or the optimal policy in those states. Alternatively, $c_t$ can be interpreted as an indicator that can be extracted from the observation $o_t$, but we made it explicit in the algorithm.

---

**Algorithm 1** Encode

---

**Inputs:** Update function $u_\theta$, encoder $q_\theta^e$, and histories $\left\{ (a_{w-1}^n, o_w^n)_{w=0}^{W-1} \right\}_{n=0}^{N-1}$.
Let $z_{-1}^n = 0$.
**for** $w = 0 \ldots W - 1$ **do**
    Let $e_{w-1}^n \sim q_\theta^e(\cdot|z_{w-1}^n, a_{w-1}^n, o_w^n)$.
    Let $z_w^n = u_\theta(z_{w-1}^n, a_{w-1}^n, e_{w-1}^n)$.
**end for**
**Returns:** $\left\{ (z_w^n, e_w^n)_{w=-1}^{W-2} \right\}_{n=0}^{N-1}$.

---

---

**Algorithm 2** Imagine

---

**Inputs:** Update function $u_\theta$, prior $q_\theta^p$, policy $g_\phi$, statistics, encoded latents and actions $\left\{(z_w^n, e_w^n, a_w^n)_{w=-1}^{W-2}\right\}_{n=0}^{N-1}$.
Let $z_{-1}^{n,w} = z_w^n$, $\hat{e}_{-1}^{n,w} = e_w^n$, $a_{-1}^{n,w} = a_w^n$.
**for** $k = 0 \ldots K - 1$ **do**
    Let $z_k^{n,w} = u_\theta(z_{k-1}^{n,w}, a_{k-1}^{n,w}, \hat{e}_{k-1}^{n,w})$.
    Let $\hat{e}_k^{n,w} \sim q_\theta^p(\cdot|z_k^{n,w}, a_k^{n,w})$.
    Let $a_k^{n,w} \sim g_\phi(\cdot|z_k^{n,w})$.
**end for**
**Returns:** $\left\{\left\{(z_k^{n,w}, \hat{e}_k^{n,w})_{k=0}^{K-1}\right\}_{w=-1}^{W-2}\right\}_{n=0}^{N-1}$.

---

**Algorithm 3** Informed Dreamer - direct reward maximization

---

**Hyperparameters:** Environment steps $S$, steps before training $F$, train ratio $R$, backpropagation horizon $W$, imagination horizon $K$, batch size $N$, replay buffer capacity $B$.

Initialize neural network parameters $\theta$, $\phi$, $\psi$ randomly, initialize empty replay buffer $\mathcal{B}$.
Let $g = 0$, $t = 0$, $a_{-1} = 0$, $r_{-1} = 0$, $z_{-1} = 0$.
Reset the environment and observe $o_0$ and $c_0$ (true at reset).
**for** $s = 0 \ldots S - 1$ **do**
    *// Environment interaction*
    Encode observation $o_t$ to $e_{t-1} \sim q_\theta^e(\cdot|z_{t-1}, a_{t-1}, o_t)$.
    Update $z_t = u_\theta(z_{t-1}, a_{t-1}, e_{t-1})$.
    Given the current history $h_t$, take action $a_t \sim g_\phi(\cdot|z_t)$.
    Observe reward $r_t$, information $i_{t+1}$, observation $o_{t+1}$ and continuation flag $c_{t+1}$.
    **if** $c_{t+1}$ is false (terminal state) **then**
        Reset $t = 0$.
        Reset the environment and observe $o_0$ and $c_0$ (true at reset).
    **end if**
    Update $t = t + 1$.
    Add trajectory of last $W$ time steps $(a_{w-1}, r_{w-1}, i_w, o_w, c_w)_{w=t-W+1}^t$ to the replay buffer $\mathcal{B}$.
    *// Learning*
    **while** $|\mathcal{B}| \geq F \wedge g < Rs$ **do**
        *// Environment learning*
        Draw $N$ trajectories of length $W$ $\left\{(a_{w-1}^n, r_{w-1}^n, i_w^n, o_w^n, c_w^n)_{w=0}^{W-1}\right\}_{n=0}^{N-1}$ uniformly from the replay buffer $\mathcal{B}$.
        Compute statistics and encoded latents

        $$\left\{(z_w^n, e_w^n)_{w=-1}^{W-2}\right\}_{n=0}^{N-1} = \text{Encode}\left(u_\theta, q_\theta^e, \left\{(a_{w-1}^n, o_w^n)_{w=0}^{W-1}\right\}_{n=0}^{N-1}\right).$$

        Update $\theta$ using $\nabla_\theta \sum_{n=0}^N \sum_{w=-1}^{W-2} L_w^n$, where $a_{-1}^n = 0$ and,
        $$L_w^n = \log q_\theta^i(i_{w+1}^n|z_w^n, e_w^n) + \log q_\theta^c(c_{w+1}^n|z_w^n, e_w^n) + \log q_\theta^r(r_w^n|z_w^n, e_w^n)$$
        $$- \text{KL}\left(q_\theta^e(\cdot|z_w^n, a_w^n, o_{w+1}^n) \| q_\theta^p(\cdot|z_w^n, a_w^n)\right).$$

        *// Behaviour learning*
        Sample latent trajectories

        $$\left\{\left\{(z_k^{n,w}, \hat{e}_k^{n,w})_{k=0}^{K-1}\right\}_{w=-1}^{W-2}\right\}_{n=0}^{N-1} = \text{Imagine}\left(u_\theta, q_\theta^p, g_\phi, \left\{(z_w^n, e_w^n, a_w^n)_{w=-1}^{W-2}\right\}_{n=0}^{N-1}\right).$$

        Predict rewards $r_k^{n,w} \sim q_\theta^r(\cdot|z_k^{n,w}, \hat{e}_k^{n,w})$, continuations flags $c_{k+1}^{n,w} \sim q_\theta^c(\cdot|z_k^{n,w}, \hat{e}_k^{n,w})$, and values $v_k^{n,w} = v_\psi(z_k^{n,w})$.
        Compute value targets using $\lambda$-returns, with $G_{K-1}^{n,w} = v_{K-1}^{n,w}$ and
        $$G_k^{n,w} = r_k^{n,w} + \gamma c_k^{n,w}\left((1-\lambda)v_{k+1}^{n,w} + \lambda G_{k+1}^{n,w}\right).$$

        Update $\phi$ using $\nabla_\phi \sum_{n=0}^{N-1} \sum_{w=-1}^{W-2} \sum_{k=0}^{K-1} G_k^{n,w}$.
        Update $\psi$ using $\nabla_\psi \sum_{n=0}^{N-1} \sum_{w=-1}^{W-2} \sum_{k=0}^{K-1} \|v_\psi(z_k^{n,w}) - \text{sg}(G_k^{n,w})\|^2$, where sg is the stop-gradient operator.
        Count gradient steps $g = g + 1$
    **end while**
**end for**

---

## D    FINAL RETURNS

We provide the final returns obtained by Dreamer and the Informed Dreamer for the Varying Mountain Hike environments in Table 1, for the Velocity Control environments in Table 2, and for the Pop Gym environments in Table 3.

Table 1: Average final return and standard deviation over five trainings in the Mountain Hike environments.

| Altitude | Random | Uninformed | Informed |
|---|---|---|---|
| False | False | $-13.70 \pm 03.32$ | $\mathbf{-13.35 \pm 02.93}$ |
| False | True | $-18.32 \pm 06.04$ | $\mathbf{-17.72 \pm 04.19}$ |
| True | False | $\mathbf{-14.78 \pm 02.44}$ | $-14.98 \pm 04.73$ |
| True | True | $-67.05 \pm 21.76$ | $\mathbf{-45.94 \pm 32.77}$ |

Table 2: Average final return and standard deviation over five trainings in the Velocity Control environments.

| Task | Uninformed | Informed |
|---|---|---|
| Acrobot Swingup | $\mathbf{113.73 \pm 108.03}$ | $112.49 \pm 54.67$ |
| Cartpole Balance | $511.60 \pm 01.95$ | $\mathbf{513.22 \pm 00.82}$ |
| Cartpole Balance Sparse | $\mathbf{491.07 \pm 00.00}$ | $485.34 \pm 49.39$ |
| Cartpole Swingup | $347.58 \pm 18.30$ | $\mathbf{371.24 \pm 05.62}$ |
| Cartpole Swingup Sparse | $36.98 \pm 42.83$ | $\mathbf{102.44 \pm 139.79}$ |
| Cheetah Run | $\mathbf{315.40 \pm 39.64}$ | $305.91 \pm 103.62$ |
| Cup Catch | $465.23 \pm 28.77$ | $\mathbf{468.32 \pm 12.53}$ |
| Finger Spin | $186.66 \pm 39.34$ | $\mathbf{245.77 \pm 61.99}$ |
| Finger Turn Easy | $359.32 \pm 76.13$ | $\mathbf{414.82 \pm 46.09}$ |
| Finger Turn Hard | $347.91 \pm 81.80$ | $\mathbf{398.38 \pm 63.40}$ |
| Hopper Hop | $91.05 \pm 29.62$ | $\mathbf{97.50 \pm 29.83}$ |
| Hopper Stand | $350.77 \pm 88.92$ | $\mathbf{384.44 \pm 74.34}$ |
| Pendulum Swingup | $\mathbf{301.01 \pm 39.80}$ | $233.66 \pm 199.66$ |
| Reacher Easy | $463.30 \pm 17.78$ | $\mathbf{477.51 \pm 14.02}$ |
| Reacher Hard | $391.94 \pm 148.99$ | $\mathbf{466.35 \pm 25.94}$ |
| Walker Run | $238.07 \pm 76.42$ | $\mathbf{271.72 \pm 63.37}$ |
| Walker Stand | $\mathbf{462.81 \pm 18.20}$ | $460.51 \pm 41.87$ |
| Walker Walk | $429.65 \pm 27.06$ | $\mathbf{440.85 \pm 49.87}$ |

Table 3: Average final return and standard deviation over give trainings in the Pop Gym environments.

| Task | Uninformed | Informed |
|---|---|---|
| Concentration | $\mathbf{00.01 \pm 00.16}$ | $-0.24 \pm 00.09$ |
| Count Recall | $-0.66 \pm 00.17$ | $\mathbf{-0.58 \pm 00.24}$ |
| Higher Lower | $\mathbf{00.39 \pm 00.07}$ | $00.31 \pm 00.12$ |
| Mine Sweeper | $\mathbf{-0.06 \pm 00.32}$ | $-0.07 \pm 00.38$ |
| Noisy Position Cart Pole | $00.21 \pm 00.19$ | $\mathbf{00.23 \pm 00.27}$ |
| Noisy Position Pendulum | $00.54 \pm 00.06$ | $\mathbf{00.55 \pm 00.05}$ |
| Position Cart Pole | $00.75 \pm 00.00$ | $\mathbf{00.75 \pm 00.00}$ |
| Position Pendulum | $00.64 \pm 00.07$ | $\mathbf{00.65 \pm 00.04}$ |
| Repeat First | $00.24 \pm 00.87$ | $\mathbf{00.56 \pm 01.00}$ |
| Repeat Previous | $-0.01 \pm 00.18$ | $\mathbf{00.44 \pm 00.13}$ |

## E    ADDITIONAL EXPERIMENTS

In this section, we provide results for non-Markovian information in the Varying Mountain Hike environments, for harder Pop Gym environments, along with the results of the flickering environments.

### E.1    NON-MARKOVIAN INFORMATION

We experiment with other levels of information in the Varying Mountain Hike environments. More precisely, we consider an information $i$ that contains an observation $\tilde{x}$ of the position $x$ (or an

observation $\tilde{y}$ of the altitude $y$) with Gaussian noise of standard deviation $\sigma_i \in [0, \sigma_o]$. In addition, in the case of environments with random orientation, we consider an information that also contains a noisy observation of the orientation $c$ replaced with a random orientation with probability $\epsilon_i \in [0, 1]$. Note that when $\sigma_i = 0$, the exact position $x$ (or altitude $y$) is encoded in the information, while when $\sigma_i = \sigma_o$, the observation $o$ is encoded in the information.

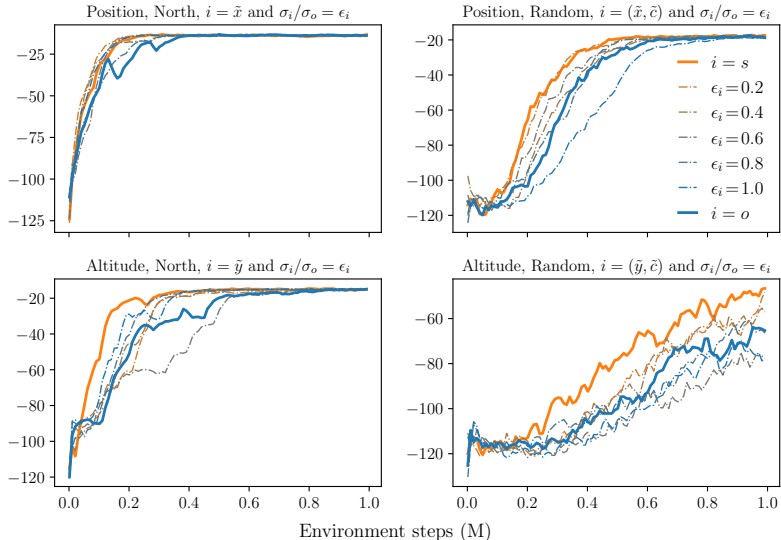

Figure 7: Varying Mountain Hike environments: average return of the Informed Dreamer with various level of information over five trainings.

As shown in Figure 7, without confidence intervals for the sake of readability, the better the information, the faster the policy converges. These results hold in all environments except that with altitude observation and fixed orientation, for which the results are more mixed. As said in Subsection 6.1, it supports the hypothesis that the more informative about the state the information is, the faster an optimal policy is learned. Moreover, it can be observed on the right in Figure 7 that when an additional information $\tilde{c}$ is not informative about the state, convergence is even slower than for the Uninformed Dreamer. This highlights again the importance of the quality of the additional information.

### E.2 HARDER POP GYM ENVIRONMENTS

Despite the optimal informed policy being equal to the optimal uninformed policy at convergence, there may exist environments for which uninformed policies do not succeed in converging to the optimal policy. One class of environments for which it seems to be the case are the environments with long time dependencies, such as the Repeat Previous environment of the Pop Gym suite. In this subsection, we study in depth this failure case of the Uninformed Dreamer for this particular environment. In the Repeat Previous environment, the agent is observing random noise, and is rewarded for outputting the observation that it got $k$ time steps ago. While in Subsection 6.3 we only considered the default Pop Gym environments, where $k = 4$ for the Repeat Previous environment, we here consider the Medium ($k = 32$) and Hard ($k = 64$) versions of this environment.

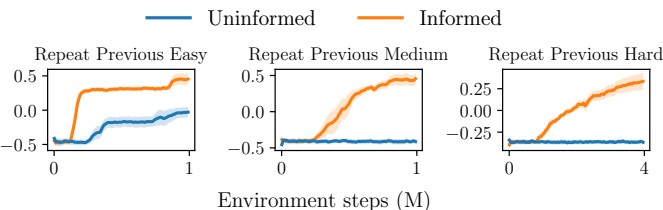

Figure 8: Uninformed Dreamer and Informed Dreamer with $i = s$ in the Repeat Previous environments: average return and standard error over five trainings.

In Figure 8, we show that the Uninformed Dreamer does not learn at all in these harder environments, while the Informed Dreamer still seems to converge towards a near-optimal policy. It once again

validates empirically the assumption that exploiting additional information about the state improves the speed of convergence towards an optimal policy. Even more, it shows that exploiting additional information about the state can lead to convergence in environments where traditional approaches fail, such as those with long time dependencies. The additional supervision provided by this Markovian information (the last $k$ observations) certainly endows the statistic $z \sim f(\cdot|h)$ with a useful encoding of the last $k$ observations, which is then decoded by the policy. Table 4 provides the final return obtained by the Uninformed Dreamer and the Informed Dreamer for these environments.

Table 4: Average final return and standard deviation over five trainings in the Repeat Previous environments.

| Task | Uninformed | Informed |
|---|---|---|
| Repeat Previous Easy | $-0.01 \pm 00.18$ | $\mathbf{00.44 \pm 00.13}$ |
| Repeat Previous Medium | $-0.41 \pm 00.06$ | $\mathbf{00.46 \pm 00.16}$ |
| Repeat Previous Hard | $-0.36 \pm 00.07$ | $\mathbf{00.33 \pm 00.19}$ |

### E.3 FLICKERING ATARI

In the Flickering Atari environments, the agent is tasked with playing the Atari games (Bellemare et al., 2013) on a flickering screen. The dynamics are left unchanged, but the agent may randomly observe a blank screen instead of the game screen, with probability $p = 0.5$. While the classic Atari games are known to have low stochasticity and few partial observability challenges (Hausknecht & Stone, 2015), their flickering counterparts have constituted a classic benchmark in the partially observable RL literature (Hausknecht & Stone, 2015; Zhu et al., 2017; Igl et al., 2018; Ma et al., 2020). Moreover, regarding the recent advances in sample-efficiency of model-based RL approaches, we consider the Atari 100k benchmark, where only 100k actions can be taken by the agent for generating samples of interaction.

For these environments, we consider the RAM state of the simulator, a $128$-dimensional byte vector, to be available as additional information for supervision. This information vector is indeed guaranteed to satisfy the conditional independence of the informed POMDP: $p(o|i, s) = p(o|i)$. Moreover, we postprocess this additional information by only selecting the subset of variables that are relevant to the game that is considered, according to the annotations provided by Anand et al. (2019). Depending on the game, this information vector might contain the number of remaining opponents, their positions, the player position, etc.

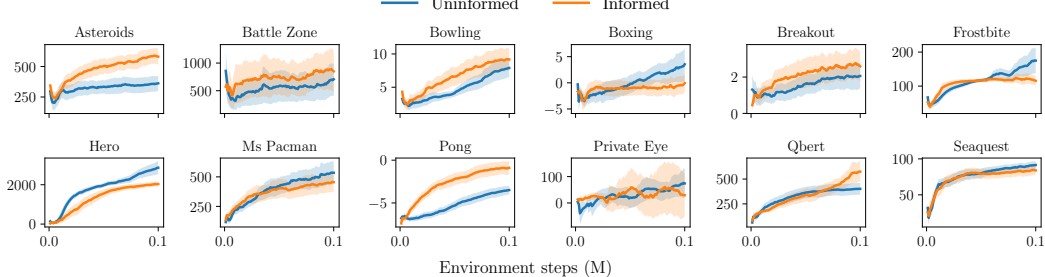

Figure 9: Uninformed Dreamer and Informed Dreamer with $i = \phi(\text{RAM})$ in the Flickering Atari environments: average return and standard error over five trainings.

Figure 9 shows that the speed of convergence and the performance of the policies is greatly improved by considering additional information for six environments, while degraded for four others and left similar for the rest. The final returns are given in Table 5, offering similar conclusions.

### E.4 FLICKERING CONTROL

In the Flickering Control environments, the agent performs one of the standard DeepMind Control tasks from images but through a flickering screen. As with the Flickering Atari environments, the dynamics are left unchanged, except that the agent may randomly observe a blank screen instead of the task screen, with probability $p = 0.5$. For these environments, we consider the state to be available as additional information, as for the Velocity Control environments.

Table 5: Average final return and standard deviation over five trainings in the Flickering Atari environments.

| Task | Uninformed | Informed |
|---|---|---|
| Asteroids | $362.17 \pm 112.95$ | $\mathbf{580.92 \pm 95.61}$ |
| Battle Zone | $706.67 \pm 776.00$ | $\mathbf{849.61 \pm 357.35}$ |
| Bowling | $07.89 \pm 02.00$ | $\mathbf{09.17 \pm 01.24}$ |
| Boxing | $\mathbf{03.54 \pm 12.33}$ | $-0.06 \pm 05.66$ |
| Breakout | $02.06 \pm 01.32$ | $\mathbf{02.59 \pm 01.47}$ |
| Frostbite | $\mathbf{174.96 \pm 84.31}$ | $115.43 \pm 30.20$ |
| Hero | $\mathbf{2864.66 \pm 1054.84}$ | $2033.51 \pm 226.50$ |
| Ms Pacman | $\mathbf{534.67 \pm 117.97}$ | $455.02 \pm 155.17$ |
| Pong | $-3.49 \pm 01.19$ | $\mathbf{-0.90 \pm 01.78}$ |
| Private Eye | $\mathbf{74.27 \pm 42.00}$ | $29.66 \pm 67.47$ |
| Qbert | $401.27 \pm 117.26$ | $\mathbf{574.70 \pm 26.92}$ |
| Seaquest | $\mathbf{91.44 \pm 13.60}$ | $83.95 \pm 21.11$ |

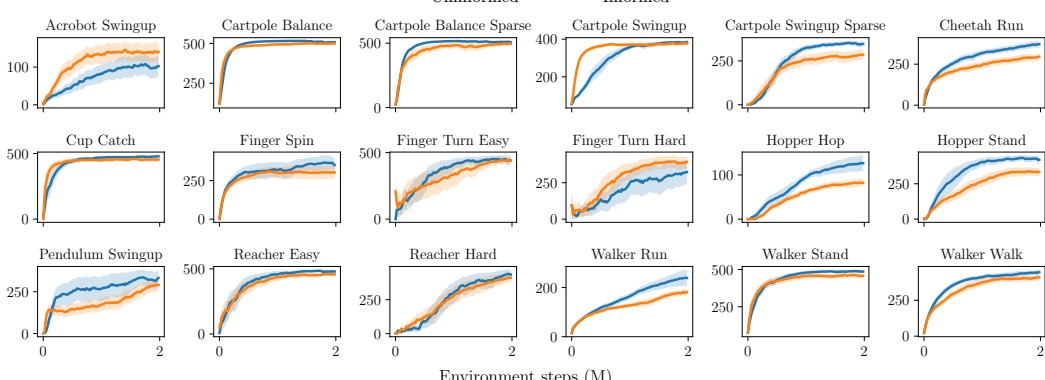

Figure 10: Uninformed Dreamer and Informed Dreamer with $i = s$ in the Flickering Control environments: average return and standard error over five trainings.

Regarding this benchmark, considering additional information seems to degrade learning, generally resulting in worse policies. This suggests that not all information is good to learn, some might be irrelevant to the control task and hinders the learning of optimal policies. The final returns are given in Table 6, and offer similar conclusions.

Table 6: Average final return and standard deviation over five trainings in the Flickering Control environments.

| Task | Uninformed | Informed |
|---|---|---|
| Acrobot Swingup | $104.87 \pm 54.88$ | $\mathbf{141.49 \pm 72.53}$ |
| Cartpole Balance | $\mathbf{508.01 \pm 00.92}$ | $499.95 \pm 24.87$ |
| Cartpole Balance Sparse | $\mathbf{507.94 \pm 03.04}$ | $495.14 \pm 69.63$ |
| Cartpole Swingup | $\mathbf{384.37 \pm 14.66}$ | $377.60 \pm 32.62$ |
| Cartpole Swingup Sparse | $\mathbf{347.07 \pm 27.63}$ | $284.53 \pm 72.05$ |
| Cheetah Run | $\mathbf{372.96 \pm 30.98}$ | $296.70 \pm 23.34$ |
| Cup Catch | $\mathbf{478.61 \pm 12.53}$ | $455.59 \pm 13.58$ |
| Finger Spin | $\mathbf{349.85 \pm 123.88}$ | $303.03 \pm 76.30$ |
| Finger Turn Easy | $\mathbf{441.53 \pm 47.13}$ | $441.16 \pm 66.91$ |
| Finger Turn Hard | $323.19 \pm 200.67$ | $\mathbf{392.48 \pm 85.25}$ |
| Hopper Hop | $\mathbf{126.72 \pm 37.89}$ | $81.92 \pm 19.90$ |
| Hopper Stand | $\mathbf{420.38 \pm 57.48}$ | $331.48 \pm 27.61$ |
| Pendulum Swingup | $\mathbf{329.35 \pm 82.31}$ | $286.53 \pm 102.18$ |
| Reacher Easy | $\mathbf{479.25 \pm 18.15}$ | $457.72 \pm 19.31$ |
| Reacher Hard | $\mathbf{433.40 \pm 214.42}$ | $412.97 \pm 27.10$ |
| Walker Run | $\mathbf{239.22 \pm 92.40}$ | $180.63 \pm 27.73$ |
| Walker Stand | $\mathbf{485.78 \pm 46.26}$ | $457.36 \pm 37.65$ |
| Walker Walk | $\mathbf{447.03 \pm 26.83}$ | $409.72 \pm 68.67$ |

