# OpenReview forum: "Informed POMDP: Leveraging Additional Information in Model-Based RL"
_ICLR.cc/2024/Conference — Submitted to ICLR 2024_

### Official Review · Reviewer_EdBW · 2023-10-29

**Soundness:** 3 good
**Presentation:** 3 good
**Contribution:** 2 fair
**Rating:** 5
**Confidence:** 3

**Summary:**

The paper introduces a new formalisation for learning optimal policies in POMDPs where the agent is allowed to additional information during training. The additional information, denoted as i, is assumed to be a function of the underlying state s, which makes the observation o conditionally independent of s given i.

The paper then proves that a representation of the history is sufficient for optimal control if it is recurrent and can accurately predict the next reward and the next information i. This is different to the prior works, which looks for sufficient representation that can predict the next reward and the next observation o.

After providing a learning objective for such sufficient representations, the paper presents a practical method that combines the learning objective and DreamerV3, a state-of-the-art MBRL method.

Through experiments, the paper investigates main two research questions. First, does the use of priviledged information during training improve the convergence of the agent? It also briefly studies the impact of different information on the speed of the agents' training.

**Strengths:**

The paper is well written.

Sufficient relevant works have been discussed.

Experimental evaluation is conducted on a diverse set of environments.

**Weaknesses:**

## Novelty

One of the main contributions of this paper is the proposal of the informed POMDP formalization.
The key novelty I find form this formalization is that it enables a new objective for learning sufficient statistics in POMDPs, which relies on predicting the next reward and information instead of the next reward and observation as in prior works.
Building on top of DreamerV3, this formalization leads to a practical MBRL algorithm that leverages additional information during training and does not need to reconstruct observations.

While I acknowledge that this is a new and promising idea, I don't find it very novel.
As discussed in the paper, asymmetric learning for learning policies in POMDPs has already been well explored.
The paper leverages this idea and combines it with MBRL approaches for POMDPs, which are not new neither.
The theoretical result is not suprising. Intuitively, if a representation of the history is predictive for the reward and information, it should also be predictive for the reward and observation by the construction of information i. The later has already been proven sufficient for optimal control by Subramanian et al. (2022).

## Experimental evaluation
I also have concerns on the experimental evaluation.

1. The authors hypothesize that leveraging this additional information will improve the convergence of the agents. However, I don't think this hypothesis is clearly supported by the results as I don't see significant improvement from informedDreamer. Moreover, for domains in which the authors claim that informedDreamer performs better at the end of training, I don't find the results very convincing due to the large standard errors. In Table 2, the large standard errors make the confidence intervals of informed and uniformed heavily overlap with each other. I would strongly suggest to run more random seeds to reduce the standard errors.

I would also like to see more reasoning for the hypothesis that leveraging such additional information will improve convergence. I disagree with the reasoning that because the information i contains more information than the observation o, the new objective will be better than the classical objective. Rather, I would argue that learning to predict i, a more complex variable, instead of o, a simpler varibale as it's function of i, might actually make the objective harder to optimize. And it is not necessary.

2. To understand the proposed method well, I think it is important to investigate the impact of different information on the training. The paper explores this question but only in one environment. I think more ablation study on this question would greatly increase the value of this research. For example, one can conduct similar controlled experiments in other domains. Or dive deeper by looking at the losses of different components of the learning objective.

**Questions:**

Questions:
- It seems that in Figure 4(b), there is no confidence interval. Does this mean that the standard error is 0?
- Does the proposed method introduce new hyperparameters? How are they tuned? For example, are there any coefficients used to balance different losses in the learning objective?

Minor:
- I would suggest to add the uninformed baseline in Figure 4(b) as well for comparison.
- Typo: Section 3.1 the discount factor \gamma \in [0,1[
- There seems to be a double citation in the second paragraph of introduction: Gregor et al. 2019.
- Figure 4 can be made larger.

---

> ### Author Response · Authors · 2023-11-20
> **Answer to Reviewer EdBW - Part 1/3**
>
> Dear Reviewer,
>
> Thank for your valuable feedback.
> We are glad to read that you found the paper promising and appreciated the novel learning paradigm.
> Below, we responds to your concerns with some clarifications, explanations or discussions that we added to the paper.
> Notably, we clarified the novelty of the paper, proposed to better discuss the hypothesis of the information improving convergence, and added additional analyses to better support this hypothesis, as requested.
> Moreover, we identified and fixed an ambiguous presentation of the result in the tables (conflicting with the figures) that may have lead you to wrong conclusions about the standard error of the mean.
>
> If these clarifications happens to address some of your concerns, or that you agree that these additional results and discussions strengthen the paper, would you be willing to raise your rating of this first step in asymmetric model-based RL?
>
> Thank you for your time and consideration.
>
> Best regards, \
> The authors.
>
> ## Novelty
>
> While we agree that asymmetric learning is not novel, it was completely left unexplored in the model-based setting.
> To the best of our knowledge, this work is indeed the first to exploit the additional information for learning a model of the environment, together with a concurrent work (Avalos et al., 2023).
> In addition, this work is the first to propose a method within this more reasonable setting of non-Markovian additional information.
> Moreover, while the theory of sufficient statistic recently developed by Subramanian et al. (2022) found a lot of interest in the partially observable RL literature, this work is the first step in extending this theory to asymmetric learning, together with a concurrent work (Sinha & Mahajan, 2023).
> Finally, this work is the first to link (symmetric) recurrent world models with the theory developed by Subramanian et al. (2022), and generalizing this to _approximate_ sufficient statistic, or approximate information state in Subramanian et al. (2022), could pave the way for deriving convergence guarantees for model-based RL in POMDP.
>
> We would also like to emphasise that the apparent simplicity of the theorem and algorithm generalizations are a direct consequence of the key conditional independence requirement, that can always be met in practice.
> Indeed, even when the eventual additional information $o^+$ does not make the observation $o$ and the state $s$ conditionally independent, it is possible to design such an information $i = (o, o^+)$ that would satisfy this independence.
> We would also like to clarify the fact that the information is not necessarily a deterministic function of the state.
>
> ## Limited performance gain
>
> We agree that the performance gain of the informed Dreamer is not as important as we might have hoped.
> However, we disagree that the result are not conclusive for several reasons.
> First, in several environments, it can be seen that the Informed Dreamer reaches the final performance of the Uninformed Dreamer in about half the number of environment steps.
> We also have conducted a new experiment, at the request of Reviewer SwZJ, in which we see that there exist environments for which the Informed Dreamer converges to a near-optimal policy while the Uninformed Dreamer completely fails to learn anything, see Appendix E.2.
> Moreover, in environments for which the Informed Dreamer converges at a higher return, the standard errors are generally lower than the order of magnitude of the difference in performance.
> Our ambiguous presentation of the tables might have mislead you in interpreting the standard deviation as the standard error of the mean in tables.
> This was a mistake not to indicate clearly what these confidence intervals represent, and we fixed it now.
> If you think that it is preferable to report the standard error of the mean in the tables, we obviously agree to modify this.
> The smaller scale of the standard error of the mean can be visualised in Figures 4 to 6, in which the confidence interval is the standard error.
> We apologise for this confusion.
>
> That being said, motivated by your remark, we also propose to better discuss the current limitations of the proposed adaptation in Section 5, and to better explain the avenues for future works in the conclusion.
> Notably, we highlight the limitations of our ELBO learning objective, in which the variational encoder is conditioned on the observation only, limiting the expressiveness of the reconstructed distribution.
> Future work may improve on this aspect by having a second encoder that is not used in the recurrence and thus not required at execution time.
> While this would come at the cost of reconstructing the observation distribution, it may also be possible to implement the latter without training the observation encoder to reconstruct the observation, but by KL-regularising its distribution to the information encoder distribution.

---

> ### Author Response · Authors · 2023-11-20
> **Answer to Reviewer EdBW - Part 2/3**
>
> ## Theoretical justification for the improved convergence speed
>
> We agree that asymmetric learning lacks theoretical motivations for exploiting the additional information, even in model-free RL.
> This is discussed extensively in Section 5.1 by Baisero & Amato (2022) and more recently in Section III.C of Sinha & Mahajan (2023).
> Despite our method being rooted in the theory of sufficient statistic (or information state) proposed by Subramanian et al. (2022), proving a faster convergence will be challenging.
> It would probably involve the notion of _approximate_ sufficient statistic, known as approximate information state in Subramanian et al. (2022), to bound the performance of the dynamic program given the error on the learned distributions.
> This would require to express the error on the observation distribution that is implicitly encoded in the information distribution, through the exploitation of the motivating inequality $I(s', i' | h, a) \geq I(s', o' | h, a)$.
> It is thus quite an ambitious program that we are considering as a future work.
>
> We also completely agree that the information could hurt learning, notably by containing irrelevant variables or exogenous variables.
> Moreover, the conditional information distribution may be more complex to approximate in practice than the observation distribution, while not being that useful to the control task.
> These reasons are now discussed in details in Section 4.
> We also want to point out that the presence of irrelevant or exogenous variables is a problem that may also concern the state / observation in symmetric RL, and that is well studied in the exogenous RL literature.
> While this body of work asks the question of what is _necessary_ in the state, the challenge in recurrent RL is still about finding a _sufficient_ representation of the history.
> Reconciling the apparent tension between these considerations is obviously a fantastic avenue for future work, but is out of the context of this article in our opinion.
>
> We also want to remind that we have shown for a simple environment in which partial observability is quite challenging (need of inferring its orientation from noisy position observations) that the speed of learning was clearly proportional to the quality of the additional information.
> This is, in our honest opinion, already a convincing empirical demonstration of our hypothesis.
> We have extended this study, as explained in the following.
>
> ## Empirical study for justifying the improved convergence speed
>
> We agree that we should have added more experiment in order to further demonstrate empirically our hypothesis.
> We thus have added some results in that direction in Appendix E.
> First, in Appendix E.2., we show for a pathological example that the Informed Dreamer can learn near-optimal in environments in which the Uninformed Dreamer is completely unable to learn, at request of Reviewer SwZJ.
> Concerning the proposition of generalizing the study about the quality of the information to other domains, we consider it to be non feasible.
> Indeed, in the other domains, we only get access to the Markovian state, and it would be difficult to artificially construct an information $i$ that provides partial information about the state but that still makes the observation conditionally independent of the state given it.
> However, we propose to generalise this to the other Mountain Hike environments, which offer very similar conclusions, see Appendix E.1.
> This constitutes in our opinion a rather convincing empirical evidence in favour our our hypothesis.
>
> ## Questions
>
> ###  No confidence interval in Figure 4(b). Does this mean that the standard error is 0?
> No it doesn't, we decided not to display them for the sake of readability. We now stated it clearly in our article.
>
> ### Does the proposed method introduce new hyperparameters?
> No, we do not introduce any new hyperparameter (the log likelihood of the observation is just replaced by the log likelihood of the information in the losses), and we kept exactly the hyperparameters as in the original Dreamer release.
>
> ## Minor
>
> ### I would suggest to add the uninformed baseline in Figure 4(b) as well for comparison.
> Excellent idea, it is done.
>
> ### Typo: Section 3.1 the discount factor \gamma \in [0,1[
> Thank you, it is fixed.
>
> ### Double citation in the second paragraph of introduction: Gregor et al. 2019.
> Thank you, it is fixed.
>
> ### Figure 4 can be made larger.
> It has now become difficult because of the additional discussions that we added to the article. But since we have considered additional environments at your request for the analyses of the quality of information, you can now consult these larger figures in Appendix E.

---

> ### Author Response · Authors · 2023-11-20
> **Answer to Reviewer EdBW - Part 3/3**
>
> ## References
>
> - Avalos, R., Delgrange, F., Nowé, A., Pérez, G. A., & Roijers, D. M. (2023). The Wasserstein Believer: Learning Belief Updates for Partially Observable Environments through Reliable Latent Space Models. In European Workshop on Reinforcement Learning.
> - Baisero, A., & Amato, C. (2022, January). Unbiased Asymmetric Reinforcement Learning under Partial Observability. In Proceedings of the International Joint Conference on Autonomous Agents and Multiagent Systems.
> - Sinha, A., & Mahajan, A. Asymmetric Actor-Critic with Approximate Information State.
> - Subramanian, J., Sinha, A., Seraj, R., & Mahajan, A. (2022). Approximate information state for approximate planning and reinforcement learning in partially observed systems. The Journal of Machine Learning Research, 23(1), 483-565.

---

> > ### Comment · Reviewer_EdBW · 2023-11-22
> >
> > Thank you for your response.
> >
> > I appreciate the value of being the first to explore asymmetric learning within MBRL. However, this aspect does not sufficiently address my concerns on the overall novelty of the approach. Also taking into account the limited performance gain of the proposed method, I choose to keep my current score.

---

> > ### Author Response · Authors · 2023-11-22
> >
> > Dear Reviewer,
> >
> > Thank you for your answer that we respect, even if we cannot agree with for these reasons:
> > - This work is the first to consider non Markovian supervision in asymmetric learning,
> > - This work is the first to extend the theory of sufficient statistics (information states) to asymmetric learning,
> > - And as you said, the first to consider MBRL for asymmetric learning.
> >
> > Moreover, the clarification resulting in a $\sqrt{5}= 2.236$ smaller standard error in the tables invalidates the argument of confidence intervals overlapping in most environments, and challenge what we believe is a too negative interpretation of the simulation results. We thus think that, together with the new results of Appendix E.1 and E.2, our results indicate that our method both significantly improves the convergence speed in many environments, and results in better final policies in many environments, especially challenging ones.
> >
> > Sincerely, \
> > The authors.

---

### Official Review · Reviewer_1Mpb · 2023-10-30

**Soundness:** 3 good
**Presentation:** 2 fair
**Contribution:** 2 fair
**Rating:** 6
**Confidence:** 4

**Summary:**

This paper proposes a model-based RL method for partially observable environments that exploits additional information during training.

The paper introduces a nice framework called the informed POMDP, which introduces an additional variable "i" (information variable) between the state and observation such that the observation is independent of the state given i.
Predicting this variable i is supposed to be easier than o --- accelerating the representation learning --- but also sufficient for optimal control (based on the fact that its sufficient to predict the observation), hence they show theoretically sound.

In practice, they do this by adjusting dreamerv3 [1] to learn to decode the state rather than the observation.
In some domains this improves the learning rate, presumably because is it easier / quicker to learn to predict the (more informative or more compactly represented) state than the observation.

Altogether I believe this is a fantastic step into a promising direction, that of exploiting additional information during training, which has been more common in model-free approaches (typically through auxiliary learning tasks, which has parallels with the proposed work).
Their dreamerv3 seems to work at least as good as the original one, fairly consistently beating it with somewhat, on domains including "mountain hike", "velocity control", and "pop gym".

[1] Hafner, D., Pasukonis, J., Ba, J., & Lillicrap, T. (2023). Mastering diverse domains through world models. arXiv preprint arXiv:2301.04104.

**Strengths:**

This paper is clearly written and proposes a solution method that should be relevant to a significant portion of the RL community: those that care about partial observability or dreamer-like solution methods.
The proposed setting, that of exploiting additional information during training in partially observable environments, is reasonable and a promising direction that has not been explored for model-based RL much yet.
Lastly, I found the formalization of the informaed POMDP and the theoretical support helpful.

So, altogether, this is a good fit for ICLR based on those reasons.

**Weaknesses:**

The main points of improvement, in my opinion, is in the actual implementation of the theoretical ideas in this paper.
In particular, the resulting algorithm is a minor change in which dreamerv3 is learned to decode the state, rather than the observation.
It is not hard to see that this, likely, will lead to an easier learning task, hence improving performance.

Furthermore, the results are not nearly as impressive as they should be for a method that suddenly assumes access to the state during training.
This is an incredibly strong assumption in most real applications and thus heavily limits its applicability.
Yet, performance-wise, we see a minor improvement on most and only one some significant performance boost.

Lastly, while the theoretical set up was nice to see and thorough, I did not find the findings particularly surprising or promising:
It is rather obvious that if predicting the observation is "good enough" than predicting anything that can fully explain (predict) the observation also has that property.

Lastly, I found it particularly frustrating how hard it was to piece together exactly the difference between the proposed method and dreamer, since the notation is just slightly different enough that it takes a lot of puzzling to align the two.

**Questions:**

N/A

---

> ### Author Response · Authors · 2023-11-20
> **Answer to Reviewer 1Mpb - Part 1/1**
>
> Dear Reviewer,
>
> Thank your for your valuable feedback.
> We are pleased to see that you appreciated our new framework and this first step in asymmetric model-based RL.
> Even if you found the algorithmic contribution minor and the generalised theorem quite trivial, it is nice to read that your appreciated that the method was supported by the theory.
> Below, we give an explanation that may address your first concern about applicability, and we provide a discussion for explaining both theoretically and practically the limited performance gain that we also propose to add to the article, along with some propositions of future works for improving this in the conclusion.
>
> If you agree that our answers and added discussions addressed some of your concerns, would you be willing to improve your rating for this work, which is both a first step in the theory of sufficient statistic for asymmetric learning and in model-based asymmetric learning?
>
> Thank you for your valuable feedback and for your consideration.
>
> Best regards, \
> The authors.
>
> ## Assumption of conditional independence and applicability
>
> While we agree that assuming state observability at training time is strong in some environments, we would like to point out that we are able to deal with any kind of additional information, which is a much more reasonable assumption, and a novelty.
> In other words, the problem that we propose to tackle is a strict generalization of the standard learning paradigm for POMDP.
> Even when the eventual additional information $o^+$ does not make the observation $o$ and the state $s$ conditionally independent, it is possible to design such an information $i = (o, o^+)$ that would satisfy this independence.
> We would also like to emphasise that the apparent simplicity of the theorem and algorithm generalizations are a direct consequence of this key conditional independence requirement, that can always be met in practice.
> Finally, to the best of our knowledge, this work is the first to propose a method within this more reasonable setting of non-Markovian additional information.
>
> ## Minor change with respect to Dreamer and limited performance gain
>
> While we agree that, when introducing this conditional independence requirement for the information, the theory motivates a very slight modification of existing algorithm, we really see this work as a first step towards developing asymmetric model-based RL methods.
> In the light of your remark, we propose to better discuss the current limitations of the proposed adaptation in Section 5, and to better explain the avenues for future works in the conclusion.
> Notably, we highlight the limitations of our ELBO learning objective, in which the variational encoder is conditioned on the observation only, limiting the expressiveness of the reconstructed distribution.
> Future work may improve on this aspect by having a second encoder that is not used in the recurrence and thus not required at execution time.
> While this would come at the cost of reconstructing the observation distribution, it may also be possible to implement the latter without training the observation encoder to reconstruct the observation, but by KL-regularising its distribution to the information encoder distribution.
>
> ## Clarifications on the difference with the original Dreamer
>
> We apologise for the important change in notation with respect to the initial Dreamer algorithm.
> We tried to find a comprise between those used in the motivating theory of Subramanian et al. (2022) and those used in the practical algorithm of Hafner et al. (2023).
> While we know that reviewers are not required to read the appendices, we believe that Appendix C may help disambiguating notations.
> Indeed, in view of this change in notations, we have written a precise pseudocode for the Informed Dreamer in Appendix C, that specifically highlight differences with the original Dreamer algorithm.
>
> ## References
>
> - Hafner, D., Pasukonis, J., Ba, J., & Lillicrap, T. (2023). Mastering diverse domains through world models. arXiv preprint arXiv:2301.04104.
> - Subramanian, J., Sinha, A., Seraj, R., & Mahajan, A. (2022). Approximate information state for approximate planning and reinforcement learning in partially observed systems. The Journal of Machine Learning Research, 23(1), 483-565.

---

> > ### Comment · Reviewer_1Mpb · 2023-11-20
> >
> > Thank you for your response. I do not disagree with your observations but I do not believe it quite addresses the concerns well enough for me to change my mind regarding the score.

---

> > > ### Author Response · Authors · 2023-11-23
> > >
> > > Thank you for your letting us know. Best regards, the authors.

---

### Official Review · Reviewer_cAA6 · 2023-11-02

**Soundness:** 3 good
**Presentation:** 3 good
**Contribution:** 3 good
**Rating:** 6
**Confidence:** 3

**Summary:**

This paper proposes informed POMDP, a formalization that utilizes additional information about the state (information beyond the agent’s observations) that is only available during training time. It is assumed that this additional training information is designed such that observation is conditionally independent of the state given this information. Using this information, the authors propose a world model, obtained by leveraging the information for learning a recurrent sufficient statistic, to sample latent trajectories. The authors then adapt Dreamer model-based RL algorithm to use the informed world model and show improvement in convergence speed when compared to Dreamer on a variety of environments.

**Strengths:**

- The informed POMDP is a natural and useful formalization that clearly articulates how additional training information can be incorporated in model-based RL. Such additional information is well motivated, especially when the agents are trained in simulation and have access to privileged information/full state information.
- The theoretical results connecting predictive models and sufficient statistics for optimal control look technically sound and are in line with prior results in similar existing work.
- The proposed approach is simple and intuitive, and can be easily adapted in many existing model-based RL approaches. The authors demonstrate this by adapting Dreamer with a modified objective and world model.
- The empirical results demonstrate clear benefits on a variety of POMDP environments when compared to Dreamer. The informed model leads to substantially faster convergence in some environments.

**Weaknesses:**

- The theoretical justification for why an informed model-based policy should converge faster, particularly in the case of informed Dreamer, isn’t completely clear. Is this solely because the recurrent state-space model in the informed world model has access to complete state information, used as the additional information, in all examples?
- While the experiments demonstrate that informed Dreamer converges faster than Dreamer in the environments tested, I don’t think this is necessarily indicative of the question of how useful the additional information is in solving POMDPs - I believe all it shows is that having access to full state information during training outperforms Dreamer in convergence speed. There should be comparison with other SOTA methods that are focused on POMDP and can exploit handle the additional information (that the Dreamer baseline doesn’t have access to in the experiments).

**Questions:**

- How sensitive are the improvements in convergence speeds to the choice of additional information? What happens when only a subset of the full state information (in addition to observations) is shared as the additional information? Do they degrade gracefully? (I acknowledge the comments on learning degradation in varying mountain hike example but this question still stands).
- Could you provide any theoretical analysis characterizing what types of additional information are most useful? Perhaps in more restricted, simpler POMDPs?
- I’m curious how consistent/different were the reconstructed observations in the case of informed world model and the baseline dreamer world model in imagined rollouts.

---

> ### Author Response · Authors · 2023-11-20
> **Answer to Reviewer cAA6 - Part 1/2**
>
> Dear Reviewer,
>
> We are pleased to read that you found our generalised formalisation of asymmetric learning natural and well motivated.
> It is also nice to read that your are enthusiastic about the established connection between asymmetric predictive models and sufficient statistics, and that your liked the simplicity of our method.
>
> We agree with your concerns on the justification for why the convergence speed would improve, and propose some explanations to address these below.
> While the theoretical motivation is indeed not satisfactory yet, we elaborate on the difficulty of establishing it but propose several avenues for future work in that direction.
> For now, we see this work as a first step in asymmetric model-based RL, as well as in the theory of sufficient statistic in asymmetric learning.
>
> If our explanations and the discussions added to the manuscript happened to address some of your concerns, would you be willing to improve your rating for the article?
>
> Thank you for your valuable review and for your consideration.
>
> Best regards, \
> The authors.
>
> ## Theoretical justification for the improved convergence speed
>
> We agree that asymmetric learning lacks theoretical motivations for exploiting the additional information, even in model-free RL.
> This is discussed extensively in Section 5.1 by Baisero & Amato (2022) and more recently in Section III.C of Sinha & Mahajan (2023).
> Despite our method being rooted in the theory of sufficient statistic (or information state) proposed by Subramanian et al. (2022), proving a faster convergence will be challenging.
> It would probably involve the notion of _approximate_ sufficient statistic, known as approximate information state in Subramanian et al. (2022), to bound the performance of the dynamic program given the error on the learned distributions.
> This would require to express the error on the observation distribution that is implicitly encoded in the information distribution, through the exploitation of the motivating inequality $I(s', i' | h, a) \geq I(s', o' | h, a)$.
> It is thus quite an ambitious program that we are considering as a future work.
> We now better discuss this compelling future work in our conclusion.
>
> We also want to remind that we have shown for a simple environment in which partial observability is quite challenging (need of inferring its orientation from noisy position observations) that the speed of learning was clearly proportional to the quality of the additional information.
> This is, in our honest opinion, already a convincing empirical demonstration of our hypothesis.
> Moreover, we extended this study in Appendix E.1 for the other Mountain Hike environments and it provides similar conclusions.
>
> ## Characterization of the usefulness of the additional information
>
> You have raised the interesting point of characterizing the usefulness of the additional information.
> It is indeed easy to see that badly designed additional information could hurt training, because the latter may contain irrelevant or exogenous information.
> Moreover, the conditional information distribution may be more complex to approximate in practice than the observation distribution, while not being that useful to the control task.
> These reasons are now discussed in details in Section 4.
> We also want to point out that the presence of irrelevant or exogenous variables is a problem that also concerns the state / observation in symmetric RL, and that is well studied in the exogenous RL literature.
> While this body of work asks the question of what is _necessary_ in the state, the challenge in recurrent RL is still about finding a _sufficient_ representation of the history.
> Reconciling the apparent tension between these considerations is obviously a fantastic avenue for future work, that we mentioned in our conclusion.
>
> ## Comparison to related works in asymmetric learning
>
> We agree that it would have been richer to gather these results of additional asymmetric learning methods in the article.
> However, these model-free RL algorithms would not benefit from the sample-efficiency of model-based RL methods such as Dreamer, and a fair comparison would be made difficult.
> Moreover, we emphasise that none of these methods were designed to deal with non Markovian additional information.
> Finally, we will not be capable of providing a fair comparison of these methods before the end of the discussion period.
>
> ## Comparison of the observation reconstructions
>
> While we agree that our article may lack more qualitative visualization of the results, other than the learning curves, we are unfortunately unable to provide this comparison.
> Indeed, one of the advantage of our method is that it does not require to reconstruct the observation but only the (hopefully more compact) information.
> Consequently, we cannot compare the quality of the reconstructions, but we known that the distributions of the observation is encoded in the distribution of information.

---

> ### Author Response · Authors · 2023-11-20
> **Answer to Reviewer cAA6 - Part 2/2**
>
> ## References
>
> - Baisero, A., & Amato, C. (2022, January). Unbiased Asymmetric Reinforcement Learning under Partial Observability. In Proceedings of the International Joint Conference on Autonomous Agents and Multiagent Systems.
> - Sinha, A., & Mahajan, A. Asymmetric Actor-Critic with Approximate Information State.
> - Subramanian, J., Sinha, A., Seraj, R., & Mahajan, A. (2022). Approximate information state for approximate planning and reinforcement learning in partially observed systems. The Journal of Machine Learning Research, 23(1), 483-565.

---

### Official Review · Reviewer_SwZJ · 2023-11-08

**Soundness:** 3 good
**Presentation:** 3 good
**Contribution:** 2 fair
**Rating:** 6
**Confidence:** 3

**Summary:**

This paper considers the problem of learning in POMDPs with privileged information during training time. The motivation is that POMDPs are in general very hard to solve; however, often during training there can be substantially more information revealed to help learn the policy than what is available at test time. This work makes progress towards this goal by proposing the Informed Dreamer algorithm which attempts to model a sufficient statistic that is enough for optimal control, combined with the model-based Dreamer algorithm. Experiments across a variety of domains are presented.

**Strengths:**

- The problem is very well motivated and I think this is relevant to people in the RL community.
- The solution is also well motivated by the theory and technically interesting from that standpoint. The method also appears to be fairly flexible to the level of privileged information that is available.
- The experiments are conducted on many different environments, which helps paint a fairly complete picture of the performance of the method.
- The paper is clearly presented.

**Weaknesses:**

- The gains are only marginally better than without privileged information. There are also no comparisons to alternative algorithms (like those mentioned in the related work), so it’s hard to judge the merits beyond how it can potentially outperform the uniformed version.
- There are a few examples of the informed method converging to a reward above the convergence of the uninformed method. There are also a few showing the opposite. Given this, I think this paper could really strengthen its position if it studied a practically interesting POMDP that would otherwise be completely intractable to solve alone (without information), but becomes solvable with training information. I believe this would constitute a very convincing result of the importance of privileged information empirically.
- The main paper does not spend much time investigating the failures that arise or trying to explain why they do. Based on the motivating theory it is not clear to me why they would happen since there is strictly more information available in the training time and the procedure would otherwise be the same. Thus, I wonder: what are the causes of informed dreamer failing to keep up with uninformed dreamer? Could it just be hyperparameters or issues with optimization? I think it would have been nice to investigate this.

**Questions:**

- Why does the reward decrease over time for some of the environments? E.g. Noisy position cart pole.
- In (10) it may be helpful to say that I is the mutual information (I assume?) to distinguish it from \tilde{I}.
- In 3.1 there’s a typo on $\gamma \in$...
- Beyond settings where $i = s$, what are practically relevant scenarios where you would see $s \rightarrow i  \rightarrow o$ non-trivially? For the sake of exposition, do you also have non-examples where you might have $i$ a training time but $s$ is not conditionally independent of $o$?

---

> ### Author Response · Authors · 2023-11-20
> **Answer to Reviewer SwZJ - Part 1/2**
>
> Dear Reviewer,
>
> We are pleased to read that you found the asymmetric learning setting well motivated, and that you appreciated the theoretical foundations as well as the flexibility of the proposed method.
> Below, we propose additional experiments and discussions to address your concerns and provide answers to your questions.
>
> If you happened to find that our additional results and explanations address your concerns and questions, would you be willing to improve your rating for this work, that in our opinion is an interesting first step both in the theory of sufficient statistic for asymmetric learning and in model-based asymmetric learning?
>
> Thank you for your valuable feedback and for your consideration.
>
> Best regards, \
> The authors.
>
> ## Comparison to related works in asymmetric learning
>
> We agree that it would have been richer to gather these additional result in the article.
> However, these algorithms would not benefit from the sample-efficiency of model-based RL methods such as Dreamer, and a fair comparison would be made difficult.
> Furthermore, we will not be capable of providing a fair comparison of these methods before the end of the discussion period.
>
> ## No clear overperformance after convergence in some environments
>
> We first want to further emphasise the partial observability of the informed policy, which means that the optimal informed policy has the same performance as the optimal uninformed policy.
> This can be clarified in the paper if you think it is worth emphasizing the focus on improving the speed of convergence.
>
> Nevertheless, we agree that beyond improving the speed of convergence, such informed objectives could permit learning in challenging environment where standard algorithms would not learn.
> First, we want to highlight that we already observed this, by obtaining a better performance after convergence in nearly all environments of the Velocity Control benchmark.
> In order to fully address your remark, we considered harder version of one of the most basic Pop Gym environments: _Repeat First_.
> In this benchmark, the agent is observing noise, and is rewarded for outputting the observation that it got $k$ time steps ago.
> While we had only considered the default _Easy_ version ($k = 4$) of the environment so far, we now considered the _Medium_ ($k = 32$) and _Hard_ ($k = 64$) versions of this environment.
> As can be seen from the results in Appendix E.2, the Informed Dreamer clearly learns near-optimal policies while the uninformed Dreamer does not learn at all.

---

> ### Author Response · Authors · 2023-11-20
> **Answer to Reviewer SwZJ - Part 2/2**
>
> ## Discussion of failures with respect to the theory and in practice
>
> We addressed your last remark by better discussing the failures that arise, both from the theory perspective in Section 4 of the article and from the practical implementation perspective in Section 5.
>
> First, we want to point out that proving a faster convergence when learning the state/information distribution instead of the observation distribution is challenging.
> Such considerations would probably involve the notion of _approximate_ sufficient statistic, known as approximate information state in Subramanian et al. (2022), to bound the performance of the dynamic program given the error on the learned distributions.
> This would require to express the error on the observation distribution that is implicitly encoded in the information distribution, through the exploitation of the motivating inequality $I(s', i' | h, a) \geq I(s', o' | h, a)$.
> It is thus a very ambitious task that we are considering as a future work.
> We now better discuss this compelling future work in our conclusion.
>
> It is more easy to explain why considering (badly designed) additional information could hurt training.
> Indeed, it is straightforward to see that the information can contain irrelevant or exogenous variables.
> Moreover, the conditional information distribution may be more complex to approximate in practice than the observation distribution, while not being that useful to the control task.
> These reasons are now discussed in details in Section 4.
> We also want to point out that the presence of irrelevant or exogenous variables is a problem that also concerns the state / observation in symmetric RL, and that is well studied in the exogenous RL literature.
> While this body of work asks the question of what is _necessary_ in the state, the challenge in recurrent RL is still about finding a _sufficient_ representation of the history.
> Reconciling the apparent tension between these considerations is obviously a fantastic avenue for future work, that we mentioned in our conclusion.
>
> We also better discuss in Section 5 the limitations of our ELBO learning objective, in which the variational encoder is conditioned on the observation only, which is another plausible explanation for the limited performance gain.
> Further work may improve on this aspect by having a second encoder that is not used in the recurrence and thus not required at execution time.
> We have made that clear in our conclusion.
>
> ## Questions
>
> ### Why does the reward decrease over time?
> There is not guarantee of monotonic improvement with such policy gradient algorithms, especially when the world model that provides the latent representations to the policy is trained jointly with the policy.
>
> ### In (10), it may be helpful to say that $I$ is the mutual information.
> Thank you, it is fixed.
>
> ### In 3.1, there's a typo in the range of $gamma$.
> Thank you, it is fixed.
>
> ### Beyond settings where $i=s$, what are practically relevant scenarios where $s \rightarrow i \rightarrow o$.
> As example for which $s \neq i$, we can consider a grasping robot for which the information are some scene variables that can be provided during training (e.g., objects types and positions) and allow to give the observation distribution but are not Markovian (e.g., objects velocities are not included).
>
> ### Do you have examples where $s$ is not conditionally independent of $o$ given the information?
> Yes, any situation in which a candidate information $\bar{i}$ (or $o^+$) is drawn independently of $o$ given $s$.
> For example when $\bar{i}$ is an additional realisation of the observation $o$, because you have the opportunity to query your sensor $O(o | s)$ several times at training time.
> In this case, the conditional independence between $o$ and $s$ given $o^+$ does not hold.
> However, as explained in Section 3 (with notation $o^+$), whatever the candidate information $\bar{i}$, the information $i = (\bar{i}, o)$ can be used and satisfies the conditional independence.
>
> ## References
>
> - Subramanian, J., Sinha, A., Seraj, R., & Mahajan, A. (2022). Approximate information state for approximate planning and reinforcement learning in partially observed systems. The Journal of Machine Learning Research, 23(1), 483-565.

---

> > ### Comment · Reviewer_SwZJ · 2023-11-22
> > **Thanks**
> >
> > Thank you. I appreciate the responses to the questions and will now discuss with the other reviewers.

---

> > > ### Author Response · Authors · 2023-11-23
> > >
> > > Thank you for your answer. Best regards, the authors.

---

### Author Response · Authors · 2023-11-20
**Note to Reviewers**

Dear Reviewers,

Once again, thank you for your valuable comments on this article.
We think that they significantly contributed to improving its clarity and soundness.

We would like to point out that some discussions are repeated in several answers, as some of you have raised similar points and we wanted each answer to be exhaustive and comprehensive.
Please find all changes and new results highlighted in red in the revised manuscript.

Best regards, \
The authors.

---

### Author Response · Authors · 2023-11-22
**End of Discussion Period**

Dear Reviewers,

As we approach the end of the discussion period, please let me know if you still have any concerns or if there are any points you would like me to clarify further.

Yours sincerely, \
The authors.

---

### Meta-Review · Area_Chair_Raxn · 2023-12-09

**Metareview:**

This paper considers the POMDP learning problem with privileged information, which provides more information to help learn the policy. An informed dreamer algorithm which attempts to model a sufficient statistic for optimal control is proposed, combined with the model-based control algorithm. Paper studied an interesting problem relevant to model-based control RL, algorithm is sound and its efficacy is justified in several benchmark experiments. while paper has novel contributions on improved sample efficiency during training,  reviewers found novelty moderately incremental (for example, comments from reviewer EdBW), the empirical improvements being marginal, also more comparisons with different SOTA baselines are needed and also issues around marginal gains when compared with non-privileged counterpart, questions about reward definitions and convergence. It's a borderline paper that is worth another revision before resubmission.

**Justification For Why Not Higher Score:**

Reviewer still pointed our several issues that the paper did not address (mentioned above). It's a borderline paper, paper is interesting but worth another revision before resubmission.

**Justification For Why Not Lower Score:**

N/A

---

### Decision · Program_Chairs · 2024-01-16

Reject